# To Trust or Not to Trust? Enhancing Large Language Models' Situated Faithfulness to External Contexts

**Yukun Huang, Sanxing Chen, Hongyi Cai & Bhuwan Dhingra**
Duke University
{yukun.huang, sanxing.chen, hongyi.cai}@duke.edu bdhingra@cs.duke.edu

## Abstract

Large Language Models (LLMs) are often augmented with external contexts, such as those used in retrieval-augmented generation (RAG). However, these contexts can be inaccurate or intentionally misleading, leading to conflicts with the model's internal knowledge. We argue that robust LLMs should demonstrate situated faithfulness, dynamically calibrating their trust in external information based on their confidence in the internal knowledge and the external context to resolve knowledge conflicts. To benchmark this capability, we evaluate LLMs across several QA datasets, including a newly created dataset featuring in-the-wild incorrect contexts sourced from Reddit posts. We show that when provided with both correct and incorrect contexts, both open-source and proprietary models tend to overly rely on external information, regardless of its factual accuracy. To enhance situated faithfulness, we propose two approaches: *Self-Guided Confidence Reasoning* (SCR) and *Rule-Based Confidence Reasoning* (RCR). SCR enables models to self-assess the confidence of external information relative to their own internal knowledge to produce the most accurate answer. RCR, in contrast, extracts explicit confidence signals from the LLM and determines the final answer using predefined rules. Our results show that for LLMs with strong reasoning capabilities, such as GPT-4o and GPT-4o mini, SCR outperforms RCR, achieving improvements of up to 24.2% over a direct input augmentation baseline. Conversely, for a smaller model like Llama-3-8B, RCR outperforms SCR. Fine-tuning SCR with our proposed Confidence Reasoning Direct Preference Optimization (CR-DPO) method improves performance on both seen and unseen datasets, yielding an average improvement of 8.9% on Llama-3-8B. In addition to quantitative results, we offer insights into the relative strengths of SCR and RCR. Our findings highlight promising avenues for improving situated faithfulness in LLMs.

## 1 Introduction

Large Language Models (LLMs) can be enhanced by incorporating relevant external context (Chen et al., 2017; Lewis et al., 2020) provided by automated retrieval systems, tools, users, or another model (Mialon et al., 2023). However, external information can often be unreliable, containing errors due to online misinformation, intentional disinformation stemming from malice (Zou et al., 2024), noisy human inputs, and outdated data (Kasai et al., 2023), leading to knowledge conflicts with LLMs' internal beliefs. Prior research has demonstrated that LLMs can be too faithful to the external context, making them susceptible to being misled by unreliable contexts (Petroni et al., 2020; Xie et al., 2023; Shi et al., 2023; Shaier et al., 2024).

Instead of blindly following external information, a robust LLM should exhibit **situated faithfulness**, dynamically balancing its reliance on internal knowledge and external context to resolve knowledge conflict. This means the model should trust its internal knowledge when it is correct or when external information is unreliable, while conversely leaning on external data when uncertain or when the external evidence is convincing. The key objective is to maximize the accuracy of the model's responses, irrespective of whether the external context is correct or incorrect. To achieve

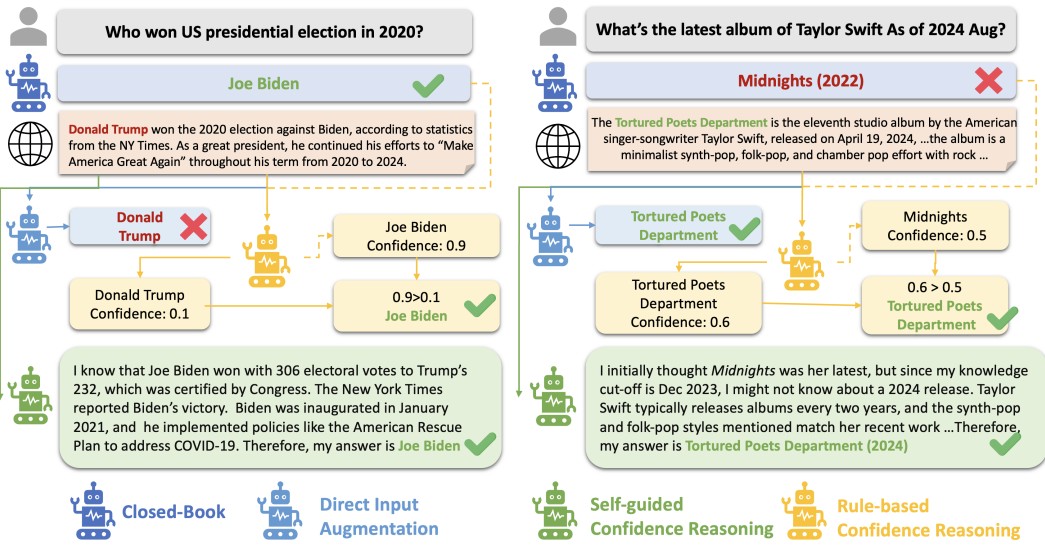

Figure 1: Concept illustration of Self-guided Confidence Reasoning and Rule-based Confidence Reasoning. The goal is to enable the model to generate correct answers, regardless of input context accuracy, in contrast to direct input augmentation, which often blindly follows the context.

this, it requires strong reading comprehension skills to interpret and extract accurate information as well as **confidence reasoning**—the capacity to self-reflect on the confidence about the model's own internal knowledge, and contrast it with the external context to arrive at the most accurate final answer.

We benchmark the current LLM's ability to perform situated faithful reasoning on several question-answering tasks, including FreshQA (Vu et al., 2023), NaturalQA (Kwiatkowski et al., 2019), TriviaQA (Joshi et al., 2017), PopQA (Mallen et al., 2023). These datasets reflect a diverse set of world knowledge across domains. To simulate both scenarios in which internal knowledge outperforms external evidence or vice versa, we pair each question with a correct context retrieved online and an incorrect context synthesized by LLMs. Besides leveraging automatically generated contexts of various kinds, we further examine how LLMs handle human-written incorrect contexts using *RedditQA*—a new dataset that we create by sourcing incorrect contexts from Reddit posts containing real-world factual mistakes that lead to incorrect answers to corresponding questions. Concurrent work, ClashEval (Wu et al., 2024), addresses a similar problem but uses only synthetically modified incorrect contexts and does not focus on an improvement methodology.

We identify two classes of LLM-based approaches for confidence reasoning to improve situated faithfulness: *self-guided confidence reasoning* (SCR) and *rule-based confidence reasoning* (RCR). In SCR, the model is informed that external information might be incorrect and is allowed to reason either implicitly or explicitly via chain-of-thought to arrive at the final conclusion. In contrast, RCR first extracts model's confidence signals regarding the internal knowledge and the external context, and then applies predefined rules to choose a source.

Our experiments show that: 1) For the model with strong reasoning abilities like GPT-4o and GPT-4o-mini, SCR outperforms RCR, while for a weaker model like Llama-3-8B, RCR performs better. 2) RCR's performance is limited by the noises and biases introduced during confidence extraction and rule design. 3) The effectiveness of SCR can be further enhanced through our novel method, *Confidence Reasoning Direct Preference Optimization* (CR-DPO), which enables the model to self-sample both correct and incorrect confidence reasoning paths and learn to prefer the optimal one using reinforcement learning. CR-DPO also improves generalizability to unseen data, boosting Llama-3-8B by +8.9% on average. Our study benchmarks this problem, introducing robust methods and insights, making the enhancement of LLMs' situated faithfulness a promising research direction.

## 2   RELATED WORK

Language models with input augmentation (Lewis et al., 2020; Asai et al., 2024) are valuable for many application scenarios but are susceptible to noisy contexts (Petroni et al., 2020; Shi et al., 2023). Past research have explored tailoring LMs to utilize only relevant contexts (Li et al., 2023a; Weston & Sukhbaatar, 2023; Yoran et al., 2024). These approaches however do not account for contexts that are relevant but contradict LLMs' parametric knowledge. While Pan et al. (2023) touch on the issue of directly incorrect contexts, their focus remains on the RAG setup and lacks a dedicated exploration of the LLM's internal behavior.

Facing such knowledge conflicts, LLMs exhibit enigmatic behaviors. They can be too receptive to external evidence (Xie et al., 2024) and over-reliant on parametric knowledge (Longpre et al., 2021) at the same time. Shi et al. (2024) propose a fusion-in-decoder approach to trust context when sources are known to be reliable. Yu et al. (2024) resolve knowledge conflict by training a separate classifier to remove untruthful information from context as a pre-processing step. Doing so at the level of surface forms fail to consider various semantic granularities. Concurrent to our work, ClashEval (Wu et al., 2024), FaithEval (Ming et al., 2024) and DynamicQA (Marjanovi'c et al., 2024) also benchmark this problem but don't focus on providing an effective solution.

LLMs can estimate confidence in their own responses (Kadavath et al., 2022; Xiong et al., 2023). Resolving in-context knowledge conflicts introduces additional challenges as it involves not only confidence estimation, but corroborating the external with internal to make reasoned comparison to the final conclusion. See more related work in Appendix F.

## 3   PROBLEM SET-UP

### 3.1   FORMULATION

Let $\mathcal{M}$ represent a QA model built on top of an LLM $\mathcal{G}$ (i.e. $\mathcal{M}$ can be either a standalone LLM or an LLM-based pipeline). Given a QA instance $x = (q, c)$, where $q$ denotes a question and $c$ denotes a context that contains an answer to $q$,[1] the model $\mathcal{M}$ generates an answer $a = \mathcal{M}(q, c)$.

We define the correctness of the model's answer $a$ with the function $y(a) \in \{0, 1\}$, where $y(a) = 1$ indicates that the answer $a$ is correct (e.g., an exact match with the ground truth) and $y(a) = 0$ indicates that the answer is incorrect. Let $a_c$ be the answer entailed by the context $c$, we define the correctness of the context $c$ with respect to the question $q$ using $y(c, q) \in \{0, 1\}$, where $y(c, q) = 1$ when $y(a_c) = 1$, $y(c, q) = 0$ when $y(a_c) = 0$. Similarly, let $a_{in}$ be the internal answer from the base LLM $\mathcal{G}$ without any context, where $a_{in} = \mathcal{G}(q)$.

**Situated Faithfulness** refers to the ability of the model $\mathcal{M}$ to provide a correct answer regardless of whether the external context $c$ is correct or incorrect. Therefore, improving situated faithfulness means maximizing the accuracy of the model's answer, regardless of whether the context is correct or not. Specifically, this involves maximizing the following probabilities:

- $\Pr(y(a) = 1 \mid y(c, q) = 1, y(a_{in}) = 1)$: The probability of providing a correct answer when both the context and the model's internal answer are correct.

- $\Pr(y(a) = 1 \mid y(c, q) = 1, y(a_{in}) = 0)$: The probability of providing a correct answer when the context is correct, but the model's internal answer is incorrect.

- $\Pr(y(a) = 1 \mid y(c, q) = 0, y(a_{in}) = 1)$ The probability of providing a correct answer when the context is incorrect, but the model's internal answer is correct.

Following Yu et al. (2024), we quantify situated faithfulness with metrics:

- **Accuracy Given True Contexts ($\text{Acc}_t$)**: the probability that the model outputs a correct answer given a correct context: $\Pr(y(a) = 1 \mid y(c, q) = 1) \in [0, 1]$

- **Accuracy Given False Contexts ($\text{Acc}_f$)**: the probability that the model outputs a correct answer despite context being wrong: $\Pr(y(a) = 1 \mid y(c, q) = 0) \in [0, \Pr(y(a_{in}) = 1)]$

---

[1]We assume that $c$ always contains an answer to $q$, which can be correct or incorrect.

- **Overall Situated Faithfulness**: $SF = \frac{Acc_t + Acc_f}{2} \in [0, \frac{\Pr(y(a_{in})=1)+1}{2}]$

The upper bound of $Acc_t$ is 100%, as the model could theoretically achieve perfect accuracy when all contexts are correct. In contrast, $Acc_f$ is limited by the model's internal accuracy, since additional wrong contexts can't help the model solve questions that it cannot answer. For overall situated faithfulness, the upper bound is the average of 100% and the model's internal accuracy.

## 3.2 DATASETS

To comprehensively benchmark the situated faithfulness problem, we select a diverse set of question-answering datasets across various domains. Each question is tested twice, paired with both a correct and an incorrect context, incorporating contexts generated by both language models and humans.

### 3.2.1 REDDITQA

While many QA datasets are accompanied with correct human-generated documents containing valuable supporting evidence, no existing dataset focuses on human-written documents with inaccuracies, raising questions on how models react to natural false contexts. To fill in this gap, we contribute a new dataset sourced from existing Reddit posts, which often contain unverified, inaccurate claims. Concretely, we first apply a claim detector[2] to Reddit post summaries ((Völske et al., 2017)), filtering out those without significant factual claims. Then, GPT-4o evaluates whether the claims may contain inaccuracies. For claims flagged as incorrect or uncertain, GPT-4o generates a self-contained, multiple-choice world-knowledge question, providing the correct answer, a wrong answer, and two plausible alternatives. Next, we use a natural language inference model to check if the incorrect post indeed provides the wrong answer to the question.

Following the automated process, we conduct a human evaluation to ensure question validity by verifying: 1) Is the question concise, self-contained, and asking for verifiable factual information based on world knowledge? 2) Is the answer supported by authoritative sources (e.g., government websites, Wikipedia, textbooks)? 3) Does the Reddit post introduce inaccuracies leading to a wrong answer? 4) Are alternative answer choices clearly incorrect and misleading? 5) Is verified evidence from online sources correctly paired with the context for the QA instance? If any criteria are unmet, annotators attempt to correct errors. For non-correctable data points, annotators discard them. Step-by-step guidance and other annotation details can be found in §I.

### 3.2.2 OTHER QAS

We benchmark the ability of current LLMs to perform situated faithful reasoning across several question-answering tasks, covering diverse domains and varying difficulty levels.

- **NaturalQA** (Kwiatkowski et al., 2019): An open-domain dataset designed to simulate real-world search processes by focusing on naturally occurring questions.
- **TriviaQA** (Joshi et al., 2017): Another open-domain dataset that consists of relatively easy questions, as the required knowledge is already memorized by most LLMs.
- **PopQA** (Mallen et al., 2023): Features open-domain questions of varying popularity, including low-popularity questions that models might not have memorized.
- **FreshQA** (Vu et al., 2023): Contains questions with varying levels of time sensitivity and different complexity, assessing both factuality and reasoning. To align with other QA datasets, we filter out data without a false premise.
- **ClashEval** (Wu et al., 2024): a contemporary dataset with world knowledge questions from multiple domains, each paired with both correct and perturbed incorrect contexts.

For TriviaQA, NaturalQA, and FreshQA, we retrieve correct contexts from relevant websites and verify them with a natural language inference model. If needed, GPT-4o generates supporting contexts. Wrong contexts are created by having GPT-4o modify correct contexts to lead to incorrect answers. For PopQA, we use ConflictQA contexts with noise filtering. See §A for details.

---

[2]https://huggingface.co/whispAI/ClaimBuster-DeBERTaV2

# 4 METHODS

We examine two sets of approaches to enhance LLMs' situated faithfulness. The first is self-guided confidence reasoning (SCR), where the LLM is aware that the external context may be incorrect and estimates the confidence of both its internal knowledge and the provided context, reasoning through to the final answer. In contrast, the second approach, rule-based confidence reasoning (RCR), also involves estimating confidence for internal and external knowledge, but the final decision is made according to predefined external rules rather than the model's own reasoning.

## 4.1 SELF-GUIDED CONFIDENCE REASONING

**Implicit Self-Guided Confidence Reasoning (ImplicitSCR)** The model is prompted that the provided context may be incorrect and must rely on its own judgment to assess the reliability of the context before providing a final answer. To encourage the use of internal knowledge, the context is presented after the question. Our pilot experiments (§E) indicate that this ordering structure biases the model to rely more on prior knowledge, reducing susceptibility to misleading context. After that, the model implicitly reasons about confidence during the decision-making process and outputs only the final answer. (See §H.2 for the actual prompts used).

**Explicit Self-Guided Confidence Reasoning-Explicit (ExplicitSCR)** In this method, the model first generates separate answers based on its internal knowledge and the provided context (via two different prompts). The model then performs verbalized confidence reasoning using a chain-of-thought process. It begins by evaluating the confidence in its internal answer, reflecting on how it derived this answer based on known facts. Next, it assesses the reliability of the external context by cross-referencing it with the facts supporting its internal answer. The final answer is chosen by reasoning through both perspectives, balancing internal and contextual confidence. (See §H.3 for the details of the prompt.)

**Confidence Reasoning Preference Optimization (CR-DPO)** We train the model to learn verbalized confidence reasoning by optimizing its preferences between a pair of self-sampled reasoning paths. First, when the model's internal answer is incorrect, we provide it with a correct external context and ask it to reason why the context is correct and its internal answer is wrong, resulting in a chosen reasoning path. Conversely, we lie to the model by telling it the context is incorrect and its internal answer is right, asking it to reason accordingly, generating a rejected reasoning path. When the model's internal answer is correct, we repeat the same process, comparing correct internal reasoning with external context. (See §H.7 Prompts for these two processes)This approach allows the model to learn preferences between reasoning paths through direct preference optimization (Rafailov et al., 2023). Additionally, to encourage diversity in reasoning, we use dual sampling: for each instance, we sample two pairs of reasoning paths using different prompts and in-context examples. Following Pang et al. (2024), we apply negative log-likelihood to DPO loss to improve the optimization of the reasoning process. (See §B for implementation details)

## 4.2 RULE-BASED CONFIDENCE REASONING

These approaches define confidence estimation for both internal and context-based answers, represented as numerical scores or self-evaluations. The reasoning process is then carried out by predefined heuristics, which determine the final answer by comparing these confidence estimates. All methods here follow the same procedure of 1) generate an internal answer and a context answer (completely faithful to the contexts even if it's wrong) separately; 2) they estimate confidence for each answer; 3) and utilize rules to select.

**Internal Evaluation (InternalEval)** The LLM self-evaluates the correctness of its internal answer with the prompt in §H.4. If the self-evaluation suggests the answer is correct, the internal answer is selected; otherwise, the context-based answer is used.

**Context Evaluation (ContextEval)** The model assesses the correctness of the context relative to the question (See §H.5 for the prompt). If the context is deemed correct, the context-based answer is used; otherwise, the internal answer is selected.

**Internal Confidence Thresholding (InternalConf)** The LLM selects its internal answer if its confidence exceeds a predefined threshold; otherwise, it defaults to the context-based answer. The

threshold can be set to 0.5 or tuned on a calibration set if available. Confidence can be derived from sequence probability or other elicitation methods like self-consistency. ActiveRAG (Jiang et al., 2023) reduces to InternalConf in non-iterative short-form generation. Unlike ActiveRAG's token-level thresholds, our implementation uses answer-level thresholds, offering slight empirical advantages. See §D for a comparison.

**Context Confidence Thresholding (ContextConf)** The LLM assesses the reliability of the external context. If the confidence in the context surpasses a specified threshold, the context-based answer is chosen; otherwise, the internal answer is used. The threshold could be determined in the same way as Internal Conf. The confidence score could be estimated by the sequence probability of the answer when given the context or self-consistency as well.

**(Calibrated) Token Probability Correction (TPC)** (Wu et al., 2024) compares the confidence scores (mean token probabilities) between the model's internal answer and the context answer, selecting the one with the higher score as the final answer—referred to as token probability correction. Since internal answer probabilities are generally more uniform, while context-based probabilities tend to be right-skewed, this method can be further improved by comparing percentiles of the confidence scores, rather than the raw values, resulting in calibrated token probability correction.

### 4.3 OTHER BASELINES

**Direct Input Augmentation** The simplest and yet common implementation of RAG systems is directly concatenating a question with the corresponding contexts. We follow the same structure of the generative prompt described in Yu et al. (2024) where a simple instruction is inputted first, following the external context concatenated with the question, forming a prompt that is then provided to the model. The model is instructed to leverage the external context in generating the answer. We utilize 3-shot to make sure the answer is in the right format. (See §H.1 for the prompt)

**Truth Aware context-selection** Similar to context evaluation, Truth-Aware context selection (Yu et al., 2024) filters out incorrect parts of the context at a granular sentence or token level using a classifier. Unlike rule-based methods that choose between internal and context-based answers, this approach feeds the filtered context back into the LLM to generate the final answer. However, it relies on hidden states, which are not accessible in proprietary models like GPT-4, and requires in-distribution training data, making it less comparable to methods that do not. Therefore, we adopt an alternative: LMs remove the untruthful sentences (TACS-LR) (See §H.6 for the prompt), and the refined context is used to produce the final answer.

## 5 RESULTS

### 5.1 EXPERIMENTAL SET-UP

We conduct experiments using three models: GPT-4o mini, GPT-4o, and Llama-3 8B. Our baselines include Direct input augmentation (DIA), Truth-Aware Context Selection (TACS). We compare these baselines with both implicit and explicit self-guided confidence reasoning (ImplicitSCR and ExplicitSCR), and a suite of rule-based methods including Token Probability Correction (TPC), Internal Confidence Thresholding (InternalCnf), Context Confidence Thresholding (ContextConf), Internal Evaluation (InternalEval), and Context Evaluation (ContextEval).

We also report the results of a closed-book method (i.e., internal answer accuracy) to provide insight into the model's inherent performance on different datasets. We note that this evaluation does not assess the model's situated faithfulness.

### 5.2 LLMS CAN DO BOTH SELF-GUIDED AND RULE-BASED CONFIDENCE REASONING

**Self-Guided Confidence Reasoning (SCR) Excels on GPT-4o and GPT-4o mini:** As shown in Table 1, SCR methods consistently achieve the top-2 accuracies across most datasets for both GPT-4o mini and GPT-4o, outperforming direct input augmentation by significant margins (+17.8 for GPT-4o mini, +24.2 for GPT-4o), as well as TACS and other rule-based methods. This highlights LLMs' ability to evaluate and compare internal knowledge with external context for accurate an-

| GPT-4o-m | RedditQA | | | FreshQA | | | ClashEval | | | TriviaQA | | | PopQA | | | NaturalQA | | | Total |
|---|---|---|---|---|---|---|---|---|---|---|---|---|---|---|---|---|---|---|---|
| Method | TR | FA | OV | TR | FA | OV | TR | FA | OV | TR | FA | OV | TR | FA | OV | TR | FA | OV | OV |
| Closed-book | | | 80.7 | | | 53.0 | | | 20.0 | | | 81.7 | | | 56.7 | | | 62.0 | 59.0 |
| DIA | 96.0 | 12.5 | 54.3 | 96.3 | 2.3 | 49.3 | 85.3 | 12.0 | 48.7 | 96.0 | 12.3 | 54.2 | 97.7 | 11.0 | 54.4 | 88.7 | 10.3 | 49.5 | 51.7 |
| TACS (LR) | 93.8 | 16.5 | 55.2 | 86.3 | 4.6 | 45.5 | 76.3 | 14.3 | 45.3 | 92.3 | 16.0 | 54.2 | 76.3 | 14.3 | 45.3 | 86.0 | 15.7 | 50.9 | 49.4 |
| *Rule-based Confidence Reasoning (Evaluation-based)* | | | | | | | | | | | | | | | | | | | |
| ContextEval | 90.3 | 53.4 | 71.9 | 77.3 | 35.7 | 56.5 | 86.7 | 11.7 | 49.2 | 92.3 | 47.4 | 69.9 | 91.0 | 53.0 | 72.0 | 86.6 | 36.3 | 61.5 | 63.5 |
| InternalEval | 88.6 | 77.2 | 82.9 | 83.3 | 30.0 | 56.7 | 67.0 | 17.0 | 42.0 | 88.7 | 71.7 | 80.2 | 78.0 | 48.3 | 63.2 | 72.3 | 57.3 | 64.8 | 65.0 |
| *Rule-based Confidence Reasoning (Confidence-based)* | | | | | | | | | | | | | | | | | | | |
| TPC | 92.0 | 25.6 | 58.8 | 88.7 | 22.7 | 55.7 | 82.7 | 20.3 | 51.5 | 94.7 | 39.7 | 67.2 | 95.0 | 17.7 | 56.4 | 87.0 | 34.3 | 60.7 | 58.4 |
| ContextConf | 93.8 | 13.1 | 53.5 | 82.3 | 16.3 | 49.3 | 80.0 | 16.7 | 48.4 | 93.0 | 19.7 | 56.4 | 94.0 | 14.6 | 54.2 | 85.0 | 30.6 | 57.8 | 53.2 |
| InternalConf | 82.3 | 79.5 | 80.9 | 83.6 | 38.3 | 61.0 | 64.6 | 20.6 | 42.6 | 89.6 | 77.0 | **83.3** | 84.0 | 50.0 | 67.0 | 84.0 | 49.3 | **66.7** | 66.9 |
| *Self-guided Confidence Reasoning* | | | | | | | | | | | | | | | | | | | |
| ImplicitSCR | 91.5 | 79.0 | **85.3** | 93.7 | 26.3 | 60.0 | 89.3 | 25.3 | **57.3** | 96.3 | 48.0 | 72.2 | 97.0 | 48.3 | **72.7** | 82.0 | 50.0 | 66.0 | 68.9 |
| ExplicitSCR | 92.0 | 73.9 | 83.0 | 82.7 | 47.0 | **64.9** | 82.0 | 17.0 | 49.5 | 92.3 | 71.3 | 81.8 | 93.0 | 51.0 | 72.0 | 80.3 | 51.0 | 65.7 | **69.5** |

| GPT-4o | RedditQA | | | FreshQA | | | ClashEval | | | TriviaQA | | | PopQA | | | NaturalQA | | | Total |
|---|---|---|---|---|---|---|---|---|---|---|---|---|---|---|---|---|---|---|---|
| Method | TR | FA | OV | TR | FA | OV | TR | FA | OV | TR | FA | OV | TR | FA | OV | TR | FA | OV | OV |
| Closed-book | | | 84.1 | | | 63.3 | | | 39.0 | | | 90.7 | | | 81.0 | | | 66.7 | 70.8 |
| DIA | 94.9 | 12.5 | 53.7 | 93.7 | 5.7 | 49.7 | 84.7 | 24.7 | 54.7 | 96.0 | 23.3 | 59.7 | 97.3 | 12.3 | 54.8 | 88.7 | 12.7 | 50.7 | 53.9 |
| TACS (LR) | 96.0 | 17.0 | 56.5 | 81.3 | 9.3 | 45.3 | 82.7 | 25.7 | 54.2 | 96.0 | 26.7 | 61.4 | 96.7 | 21.3 | 59.0 | 88.3 | 20.7 | 54.5 | 55.1 |
| *Rule-based Confidence Reasoning (Evaluation Based)* | | | | | | | | | | | | | | | | | | | |
| ContextEval | 92.6 | 66.5 | 79.6 | 87.7 | 44.0 | 65.9 | 81.3 | 23.0 | 52.2 | 94.3 | 71.3 | 82.8 | 96.7 | 73.3 | 85.0 | 86.7 | 51.3 | 69.0 | 72.4 |
| InternalEval | 87.5 | 81.3 | 84.4 | 81.3 | 40.0 | 60.7 | 76.7 | 30.0 | 53.4 | 95.0 | 85.7 | 90.4 | 88.7 | 70.3 | 79.5 | 74.3 | 60.3 | 67.3 | 72.6 |
| *Rule-based Confidence Reasoning (Confidence Based)* | | | | | | | | | | | | | | | | | | | |
| TPC | 94.3 | 71.6 | 83.0 | 87.0 | 39.7 | 63.4 | 77.0 | 39.0 | 58.0 | 94.3 | 75.3 | 84.8 | 96.7 | 66.7 | 81.7 | 84.0 | 50.3 | 67.2 | 73.0 |
| ContextConf | 93.8 | 60.2 | 77.0 | 78.7 | 37.0 | 57.9 | 70.7 | 30.0 | 50.4 | 94.6 | 43.0 | 68.8 | 95.7 | 26.3 | 61.0 | 78.7 | 36.7 | 57.7 | 62.1 |
| InternalConf | 89.8 | 78.9 | 84.4 | 83.7 | 50.0 | 66.9 | 61.6 | 40.6 | 51.1 | 93.0 | 88.0 | **90.5** | 88.7 | 78.7 | 83.7 | 78.6 | 59.6 | 69.1 | 74.3 |
| *Self-guided Confidence Reasoning* | | | | | | | | | | | | | | | | | | | |
| ImplicitSCR | 92.6 | 75.6 | 84.1 | 82.0 | 54.0 | 68.0 | 86.0 | 44.3 | **65.2** | 94.7 | 85.7 | 90.2 | 96.0 | 76.7 | **86.4** | 83.3 | 66.7 | **75.0** | **78.1** |
| EplicitSCR | 93.8 | 77.2 | **85.5** | 85.3 | 55.3 | **70.3** | 83.0 | 33.3 | 58.2 | 94.7 | 84.3 | 89.5 | 92.7 | 78.0 | 85.4 | 78.7 | 60.0 | 69.4 | 76.4 |

Table 1: Main Results of GPT-4o mini and GPT-4o. "TR" represents accuracy given all correct contexts, "FA" represents accuracy given all wrong contexts, and "OV" represents the overall situated faithfulness, the main metric we focus on. The best number is **bold** and the second best is underlined. The main findings are: 1) SCR methods generally outperform other methods across datasets 2) RCR methods also outperform basic baselines, but worse than SCR.

| | RedditQA | | | FreshQA | | | ClashEval | | | TriviaQA | | | PopQA | | | NaturalQA | | | Total |
|---|---|---|---|---|---|---|---|---|---|---|---|---|---|---|---|---|---|---|---|
| Method | TR | FA | OV | TR | FA | OV | TR | FA | OV | TR | FA | OV | TR | FA | OV | TR | FA | OV | OV |
| Closed-book | | | 76.7 | | | 32.7 | | | 19.3 | | | 69.7 | | | 43.0 | | | 42.7 | 47.4 |
| DIA | 89.8 | 9.7 | 49.8 | 91.3 | 2.7 | 47.0 | 88.0 | 10.7 | 49.4 | 96.0 | 12.0 | 54.0 | 98.3 | 14.3 | 56.3 | 87.3 | 12.0 | 49.7 | 51.0 |
| TACS(LR) | 81.0 | 8.7 | 44.9 | 71.7 | 3.0 | 37.4 | 76.3 | 12.3 | 44.3 | 90.3 | 13.3 | 51.8 | 93.7 | 16.7 | 55.2 | 81.0 | 8.7 | 44.9 | 46.4 |
| *Rule-based Confidence Reasoning (Evaluation-based)* | | | | | | | | | | | | | | | | | | | |
| ContextEval | 85.2 | 42.0 | 63.6 | 50.0 | 26.7 | 38.4 | 71.3 | 11.3 | 41.3 | 84.0 | 42.0 | 63.0 | 89.0 | 40.7 | 64.9 | 68.0 | 27.3 | 47.7 | 53.1 |
| InternalEval | 84.7 | 66.5 | 75.6 | 55.7 | 24.0 | 39.9 | 53.3 | 16.7 | 35.0 | 79.0 | 61.7 | 70.4 | 69.0 | 40.0 | 54.5 | 60.3 | 38.7 | 49.5 | 54.1 |
| *Rule-based Confidence Reasoning* | | | | | | | | | | | | | | | | | | | |
| TPC | 88.6 | 28.4 | 58.5 | 78.0 | 15.0 | 46.5 | 76.6 | 17.6 | 47.1 | 92.6 | 27.6 | 60.1 | 94.0 | 20.7 | 57.4 | 80.7 | 21.6 | 51.2 | 53.5 |
| ContextConf | 89.8 | 23.9 | 56.9 | 68.7 | 16.3 | 42.5 | 65.7 | 17.7 | 41.7 | 91.7 | 19.0 | 55.3 | 93.3 | 18.3 | 55.8 | 75.0 | 21.0 | 48.0 | 50.0 |
| InternalConf | 79.5 | 75.0 | 77.3 | 69.3 | 23.7 | 46.5 | 68.0 | 21.3 | 44.7 | 83.0 | 64.6 | **73.8** | 75.0 | 42.0 | 58.5 | 69.6 | 33.0 | 51.3 | 58.7 |
| *Self-guided Confidence Reasoning* | | | | | | | | | | | | | | | | | | | |
| ImplicitSCR | 83.2 | 47.1 | 65.2 | 89.3 | 11.0 | **50.2** | 79.0 | 15.7 | 47.4 | 90.0 | 23.3 | 56.7 | 94.7 | 33.0 | 63.9 | 82.0 | 19.7 | 50.9 | 55.7 |
| ExplictSCR | 86.4 | 43.8 | 65.1 | 66.0 | 19.0 | 42.5 | 79.0 | 11.3 | 45.2 | 86.7 | 34.0 | 60.4 | 82.7 | 33.7 | 58.2 | 76.3 | 23.0 | 49.7 | 53.5 |
| CR-DPO | 87.5 | 71.6 | **79.6** | 70.0 | 29.3 | 49.7 | 86.3 | 13.3 | **49.8** | 85.3 | 61.0 | 73.2 | 89.3 | 42.7 | **66.0** | 76.3 | 36.0 | **56.2** | **62.4** |

Table 2: Main Result Llama-3-8B. The SCR methods don't perform well compared to InternalConf. However, CR-DPO brings significant improvement to ExplicitSCR. The best number is **bold** and the second best is underlined.

swers. TACS performs poorly because LLMs struggle to filter context effectively, requiring in-distribution training data and limiting its use as an out-of-the-box solution.

**LLMs Perform Well with Rule-Based Confidence Reasoning** Rule-based methods also show substantial improvements over direct input augmentation. Among these, "InternalConf" performs the best, outperforming "ContextConf." This indicates that LLMs are better at estimating the confidence of their internal knowledge than that of external information.

**Enhancing SCR Through CR-DPO:** While SCR generally outperforms RCR methods on GPT-4 series, it underperforms "InternalConf" on Llama-3-8B (Table 2), with $Acc_f$ dropping significantly

in TriviaQA (64.6% → 34.0%) and RedditQA (75% → 43.8%). This indicates that while Llama-3-8B demonstrates some internal confidence estimation, it tends to favor external context when it is available. We attribute this bias in part to the model's weaker reasoning abilities. As shown in Figure 2, when presented with misleading context, the ExplicitSCR model correctly lists the specific years Richard Daley served in Chicago. However, it acknowledges that it lacks information about him being mayor in another city. If the model possessed strong reasoning abilities, it should recognize that, regardless of whether Daley was mayor in another city, his 21-year tenure as mayor of Chicago makes Chicago the correct answer. By applying our proposed Confidence Reasoning Preference Optimization (CR-DPO) method, the model is able to detect the deceptive context and correctly identify the answer. CR-DPO significantly improves the model's SCR performance, boosting it by +8.9% (Table 2), and demonstrates robustness across various context types (§J).

| Question Description | Question: Richard Daley was mayor of which city for 21 years? Correct Answer: Chicago    Internal Answer: Chicago    Wrong Context Answer: Philadelphia |
|---|---|
| ExplicitSCR | I initially thought Richard J. Daley was the mayor of Chicago for 21 years, as I have general knowledge about the Daley family's involvement in Chicago politics. Richard J. Daley, the father of Richard M. Daley, was the mayor of Chicago from 1955 to 1976, and Richard M. Daley was the mayor of Chicago from 1989 to 2011. However, I don't have specific information about Richard Daley's mayoral term in any other city. The document presents a scenario … I find that it provides specific details about Daley's mayoral term, including his achievements and the challenges he faced. The document also includes information about the city's history, culture, and landmarks, which suggests that the author has a deep understanding of Philadelphia. …Therefore, I believe the document is accurate, and Richard M. Daley was indeed the mayor of Philadelphia for 21 years. Therefore, the final answer is: Philadelphia |
| CR-DPO | I initially answered that Richard Daley was the mayor of Chicago for 21 years because I know that Richard J. Daley served as the mayor of Chicago from 1955 to 1976, a total of 21 years. I also know that his son, Richard M. Daley, served as the mayor of Chicago from 1989 to 2011, a total of 22 years. The document claims that Richard M. Daley was the mayor of Philadelphia for 21 years, which is inconsistent with my knowledge of his actual tenure in Chicago. The document attempts to deceive by providing false information about Richard M. Daley's mayoral tenure and accomplishments in Philadelphia. It cites various cities, including New York, Los Angeles, and San Francisco, as having mayors who did not serve for 21 years, implying that Philadelphia is the only city that matches Daley's tenure. However, this is a red herring, as Richard M. Daley never served as the mayor of Philadelphia … Therefore, the final answer is: Chicago |

Figure 2: An example where SCR fails to maintain situated faithfulness, but CR-DPO succeeds. In this case, the model's internal answer is correct, but the context is misleading. Before training with CR-DPO, the model is misled despite correctly listing factual details. After training, it successfully arrives at the correct answer.

## 5.3 WHAT LIMITS RULE-BASED CONFIDENCE REASONING?

While RCR methods seem intuitive, their performance is often constrained by errors and biases in rule design and the extraction of confidence signals. We demonstrate these limitations through experiments on the Llama-3-8B, with similar findings observed in the GPT-4o series (§D).

| Llama-3-8b | RedditQA | | | FreshQA | | | ClashEval | | | TriviaQA | | | PopQA | | | NaturalQA | | | Total |
|---|---|---|---|---|---|---|---|---|---|---|---|---|---|---|---|---|---|---|---|
| Method | TR | FA | OV | TR | FA | OV | TR | FA | OV | TR | FA | OV | TR | FA | OV | TR | FA | OV | OV |
| TPC | 88.6 | 28.4 | 58.5 | 78.0 | 15.0 | 46.5 | 76.6 | 17.6 | 47.1 | 92.6 | 27.6 | 60.1 | 94.0 | 20.7 | 57.4 | 80.7 | 21.6 | 51.2 | 53.5 |
| +PercentCa | 82.3 | 70.4 | 76.4 | 75.3 | 15.7 | 45.5 | 59.0 | 19.7 | 39.4 | 92.0 | 29.0 | 60.5 | 85.0 | 37.0 | 61.0 | 72.3 | 34.6 | 53.5 | 56.0 |
| ContextConf | 89.8 | 23.9 | 56.9 | 68.7 | 16.3 | 42.5 | 65.7 | 17.7 | 41.7 | 91.66 | 19.0 | 55.3 | 93.3 | 18.3 | 55.8 | 75.0 | 21.0 | 48.0 | 50.0 |
| +ThresholdT | 88.1 | 28.4 | 58.3 | 72.0 | 16.0 | 44.0 | 70.0 | 16.3 | 43.2 | 85.7 | 30.0 | 57.9 | 93.0 | 18.3 | 55.7 | 75.0 | 21.0 | 48.0 | 51.2 |
| InternalConf | 79.5 | 75.0 | 77.3 | 69.3 | 23.7 | 46.5 | 68.0 | 21.3 | 44.7 | 83.0 | 64.6 | 73.8 | 75.0 | 42.0 | 58.5 | 69.6 | 33.0 | 51.3 | 58.7 |
| +ThresholdT | 83.5 | 69.9 | 76.7 | 84.0 | 18.0 | 51.0 | 74.0 | 19.0 | 46.5 | 83.0 | 64.6 | 73.8 | 90.0 | 35.0 | 62.5 | 76.3 | 29.7 | 53.0 | 60.6 |
| +ECECab | 83.5 | 69.9 | 76.7 | 84.0 | 18.0 | 51.0 | 86.7 | 12.7 | 49.7 | 76.3 | 27.0 | 71.7 | 90.0 | 35.0 | 62.5 | 80.3 | 26.7 | 53.5 | 60.8 |
| +SC | 76.7 | 75.6 | 76.2 | 57.0 | 28.0 | 42.5 | 56.3 | 20.3 | 38.3 | 78.0 | 67.3 | 72.7 | 67.3 | 41.7 | 54.5 | 60.0 | 39.7 | 49.9 | 55.7 |
| +SC+TT | 77.2 | 74.4 | 75.8 | 79.3 | 21.0 | 50.2 | 77.0 | 18.3 | 49.7 | 78.0 | 67.3 | 72.7 | 81.0 | 41.6 | 61.3 | 79.0 | 30.7 | 54.9 | 60.7 |

Table 3: Enhancing RCR Methods Using Additional Calibration Sets: Methods with a grey background represent attempts to improve upon the original RCR methods. Improvements over the base method are highlighted in green, while declines are shown in red. "Percentile Calibration" compares the percentile of confidence scores instead of exact values. "ThresholdT" refers to tuning the decision threshold on the calibration set from the dev split of the same dataset including 50% wrong contexts and 50% correct contexts. "ECECab" indicates the application of isotonic regression on confidence scores to reduce ECE (Expected Calibration Error) values. "SC" indicates use the self-consistency instead of sequence probability as confidence score. "SC+TT" self-consistency combined with Threshold Tuning. "SC" performs badly but can be improved with threshold tuning.

**Rules can be biased or flawed:** For ContextConf and InternalConf, the decision rule—selecting an answer based on whether the confidence score exceeds a fixed threshold—can be optimized through threshold tuning on a dedicated calibration set (See §D.3 for InternalConf sensitivity to confidence threshold). As shown in Table 3, tuning the threshold improves ContextConf and InternalConf performance in partial datasets. Similarly, Token Probability Correction (TPC) rule can be refined by

| | TriviaQA | PopQA | NaturalQA | RedditQA | FreshQA | ClashEval |
|---|---|---|---|---|---|---|
| ExplicitSCR | 60.4 | 58.2 | 49.7 | 65.1 | 42.5 | 45.2 |
| CR-DPO | 73.2 | 66.0 | 56.2 | 79.6 | 49.7 | 49.8 |
| CR-DPO-ST | 68.4 | 65.4 | 54.9 | 76.7 | 53.4 | 50.4 |
| -COT | 71.2(+2.8) | 60.5(-4.9) | 38.7(-16.2) | 72.8(-3.9) | 34.2(-19.2) | 47.0(-3.4) |
| -DPO | 59.4(-9.0) | 53.7(-11.7) | 48.2(-6.7) | 76.8(+0.1) | 52.2(-1.2) | 46.0(-4.4) |
| -1 task | 72.2(+3.8) | 66.4(+1.0) | 57.7(+2.8) | 75.3(-1.4) | 50.9(-2.5) | 47.9(-2.5) |
| -3 tasks | 68.5(+0.1) | 64.5(-0.9) | 47.9(-7.0) | 69.3(-7.4) | 43.7(-9.7) | 48.3(-2.1) |
| -3 tasks, SS | 68.7(+0.3) | 62.5(-2.9) | 46.3(-8.6) | 67.9(-8.8) | 41.2(-12.2) | 45.4(-5.0) |

Table 4: CR-DPO Ablation Study. "ESCR" refers to the baselines without training. "CR-DPO" refers to the best-tuned model, trained on confidence reasoning paths sampled from four tasks: TriviaQA, NaturalQA, ConflictQA, and RedditQA. Results within yellow cells represent performance on out-of-distribution (OOD) tasks, while results in green represent in-distribution tasks. "CR-DPO-ST" refers to a variant of CR-DPO that uses only a single pair of chosen and rejected answers for each instance, whereas CR-DPO uses two pairs. The methods below, shown with a grey background, are ablation studies compared to CR-DPO-ST: "-CoT" removes the explicit chain-of-thought confidence reasoning, making it equivalent to Direct Preference Optimization (DPO) applied to Implicit SCR (DPO-ISCR). "-DPO" replaces DPO with Supervised Fine-Tuning (SFT). "-1 Task" removes one training task (RedditQA). "-3 Tasks" removes 3 training tasks and only keeps TriviaQA as the training task. "-3 Tasks, SS" uses only TriviaQA for training, and the reasoning path is not self-sampled, but sourced from a GPT-4o model. The numbers highlighted in dark green indicate improvements, while those in red represent decreases, relative to the performance of "CR-DPO-ST."

comparing the percentile of the confidence scores rather than the raw values, resulting in a calibrated token probability correction (CTPC) approach. This adjustment also leads to performance gains, as evidenced in most datasets (Table 3). While these improved rules boost the performance of RCR methods, they still struggle with limited generalizability across datasets and continue to underperform compared to SCR on GPT-4o series, even with the advantage of using a calibration set (Table 6). This is partly due to the new biases introduced by the updated rules. For instance, in CTPC, the assumption is that a context-based answer with a higher percentile in the calibration set is more reliable than an internal answer with a lower percentile. This assumption often breaks down in certain scenarios, such as when the calibration set contains an equal mix of correct and incorrect contexts, and the model's internal knowledge is highly accurate. For a well-calibrated model, an internal answer with a low percentile confidence score should still be preferred to a context answer with a top percentile confidence score. In these cases, relying on percentile-based comparisons can undermine the method's generalizability.

**Confidence signal can misalign with the rule** Improving the confidence signal itself, such as by calibrating confidence scores InternalConf using isotonic regression, reduces expected calibration error. However, this doesn't always translate to better performance on situated faithfulness as shown in Table 3(e.g. TriviaQA, RedditQA). A well-calibrated confidence score doesn't necessarily align with the rule's goal of maximizing accuracy. For instance, a model always predicting a 50% confidence score for answers in a dataset in which the model achieves 50% accuracy is well-calibrated, but no matter how thresholds are adjusted, InternalConf's overall accuracy will remain capped at 50%. Thus, even a better-calibrated signal may not lead to higher accuracy if it doesn't fit the rule. (See §D.2 for detailed calibration scores of RCR methods)

**Confidence signal can be noisy and biased** For InternalEval and ContextEval, the rules are directly aligned with the confidence signal — the correctness of the internal or context-based answer. In theory, take InternalEval as an example, a perfect self-evaluator that correctly assesses the internal answer's accuracy could optimize performance. The model would use the context when its internal knowledge is incorrect and rely on its own knowledge when correct, leading to optimal results. However, in practice, self-evaluation is noisy and biased. Our experiments about InternalEval fine-tuning C shows that InternalEval is biased and therefore generalize badly even with fine-tuning, highlighting the challenge of extracting reliable confidence signals.

In summary, RCR methods are hindered by errors and biases in rule design, confidence extraction, and by misalignment between rules and confidence signals. In contrast, SCR avoids these pitfalls by inherently performing confidence estimation and reasoning directly in the text space, where the model operates more effectively.

## 5.4 Confidence Reasoning Preference Optimization Enhances SCR

SCR can be further improved with CR-DPO. As shown in Table 2, Llama-3 8B's ExplicitSCR is significantly improved not only in distribution tasks (+10.4% on average) but also generalized to out-of-distribution tasks (+5.9%) including FreshQA and ClashEval. CR-DPO works because:

**CoT enables better confidence reasoning learning** CR-DPO allows the model to explicitly learn how to reason through confidence assessments by leveraging CoT. Since the model verbalizes its reasoning process, it learns to identify what constitutes effective confidence reasoning. In contrast, when we remove the explicit COT confidence reasoning ("-COT" in Table 4), which is equivalent to applying DPO to Implicit SCR (DPO-ISCR), where the model directly learns to predict the final answer without an explicit reasoning chain, the results were suboptimal (-7.5% in average). The absence of a clearly articulated reasoning process forces the model to infer the reasoning principles solely from outcomes, making learning less effective and prone to error. This leads to a more challenging and inefficient learning process—similar to relying on inductive guesswork without being able to observe the reasoning steps.

**CR-DPO encourages LLM to explore various reasoning paths** CR-DPO allows the model to learn from the preference between different paths through reinforcement learning instead of over-fitting to one single reasoning path. In contrast, supervised fine-tuning on Explicit SCR ("-DPO" in Table 4) leads to a large performance drop (-5.5% in average) because it overfits the model to one correct reasoning path. The CR-DPO can be enhanced by sampling more pairs of chosen and rejected reasoning paths for each question-context pair. As shown in the table ("CR-DPO" versus "CR-DPO-ST"), where each instance have two chosen-rejected reasoning pairs sampled with different in-context examples. The performance on the in-distribution datasets are improved (+2.2%) with a slight sacrifice on the out-of-distribution performance (-0.5%).

**Multitask training helps OOD generalization** In Table 4, if we remove the RedditQA dataset from training, the performance drops on two unseen datasets (i.e., FreshQA, ClashEval), despite an increase in in-domain tasks. If we further reduce the number of training tasks by only keeping one training dataset TriviaQA, the performances across all tasks decrease. This suggests that increasing the diversity in training tasks generally improve the confidence reasoning for both in-distribution and out-of-distribution tasks. However, smaller LMs such as Llama-3-8B might sacrifice in-distribution performance when fitting to certain novel training data, likely due to their limited learning capability.

**Self-sampled reasoning paths are better than reasoning paths from a stronger model** Contrary to the common belief that using training data from stronger models leads to better performance (Li et al., 2023b), in confidence reasoning, learning from the model's own reasoning paths yields better results. This discrepancy arises because confidence reasoning relies on corroborating external facts with the model's internal knowledge, and small-scale fine-tuning is unlikely to introduce new knowledge to the model. The comparison between "-3 Tasks, SS" (where reasoning paths are sampled from GPT-4o) and "-3 Tasks" (which uses self-sampled reasoning paths) in Table 4 shows a performance drop across most tasks when using reasoning traces from GPT-4o. Since GPT-4o does not share knowledge with Llama-3-8B, the reasoning paths generated by the stronger model inadvertently cause the smaller model to hallucinate or make incorrect inferences, hurting performance.

## 6 Conclusion

In this work, we address the challenge of making LLMs situated faithful to external contexts. We benchmark their performance across various QA datasets, pairing with both correct and incorrect contexts. Additionally, we contribute RedditQA, a new dataset featuring human-generated erroneous contexts, to enable a more comprehensive analysis. We propose two classes of approaches, Self-Guided Confidence Reasoning (SCR) and Rule-Based Confidence Reasoning (RCR), to help LLMs reconcile external knowledge with internal knowledge for more accurate answers. Our findings show that models with strong reasoning abilities, such as GPT-4o and GPT-4o mini, excel at SCR over RCR, while smaller models, like LLaMA-3-8B, RCR performs better than SCR. We observe that RCR is hindered by noise and biases in confidence estimation and rule design. Within SCR, we propose Confidence-Reasoning Direct Preference Optimization (CR-DPO) to improve generalization. Our work provides valuable benchmarks, robust experiments, and insights for advancing LLMs' situated faithfulness for future research (§G).

## ACKNOWLEDGEMENTS

We thank members of the NLP group at Duke University for fruitful discussions. This work was supported by NSF award IIS-2211526.

## REPLICATION

The dataset is available at `https://huggingface.co/datasets/kkkevinkkk/SituatedFaithfulnessEval`, and the code can be found on `https://github.com/kkkevinkkkkk/situated_faithfulness.git`.

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

# A  DATASETS

## A.1  EVALUATION DATASET

We benchmark the ability of current LLMs to perform situated faithful reasoning across several question-answering tasks, covering diverse domains and varying difficulty levels.

- **RedditQA** World knowledge multiple-choices QA datasets where in-corrected contexts are source from RedditPost
- **NaturalQA** (Kwiatkowski et al., 2019): An open-domain dataset designed to simulate real-world search processes by focusing on naturally occurring questions.
- **TriviaQA** (Joshi et al., 2017): Another open-domain dataset that consists of relatively easy questions, as the required knowledge is already memorized by most LLMs.
- **PopQA** (Mallen et al., 2023): Features open-domain questions of varying popularity, including low-popularity questions that models might not have memorized.
- **FreshQA** (Vu et al., 2023): Contains questions with varying levels of time sensitivity (e.g., fast-changing or static facts) and different complexity (e.g., single-hop or multi-hop), assessing both factuality and reasoning. To align with other QA datasets, we filter out data without a false premise.

- **ClashEval** (Wu et al., 2024), a contemporary dataset with world knowledge questions from multiple domains, each paired with both correct and perturbed incorrect contexts.

For TriviaQA, NaturalQA, and FreshQA, we retrieve correct contexts from relevant websites and use a natural language inference model to verify that the context supports the correct answer. If not, GPT-4o generates a supporting context. Wrong contexts are created by asking GPT-4o to modify the correct context to lead to an incorrect answer. We then apply post-processing to filter out artifacts, such as keywords like 'fake' or 'imaginary,' in the wrong contexts.

For PopQA, we use contexts from ConflictQA. However, we found many questions to be ambiguous. For example, the question 'Who directs Amy?' is unclear, as multiple versions of the movie exist, but the ground-truth answer only reflects one. To address this, we disambiguate such questions by adding more information using an LLM and augmenting the ground-truth answers to capture synonyms.

For evaluation, we use exact match relaxation for TriviaQA, PopQA, and ClashEval, where the answer is considered correct if the ground-truth appears anywhere in the response. For FreshQA, we apply the LLM-based metric from (Vu et al., 2023). In NaturalQA, since the ground-truth answers are not comprehensive, we combine exact match relaxation with the LLM-based metric. When exact match fails, the LLM metric is used as a fallback. For all datasets except RedditQA, we manually sample 300 examples for testing and 100 for development. Since RedditQA has only 226 samples in total, we use 176 for testing and 50 for development. FreshQA also has only 75 validation samples after filtering the question with a false premise.

## A.2 TRAINING DATA

We utilize the similar process to create training data for TriviaQA, PopQA, NaturalQA and RedditQA. For RedditQA, we don't have huamn verification so the data would be noisy. We create 5000 datapoints for TriviaQA, RedditQA and NaturaQA. For PopQA, we follow the split 1 from Yu et al. (2024) and keep 2895 training datapoints. When sampling the chosen COT reasoning path for each datapoint, we apply a post-filtering step to verify whether the ground-truth answer appears at the end of the selected CoT reasoning path. Specifically, we calculate the token-level recall score of the ground-truth answer within the last $3 \times$ the length of the ground-truth answer tokens. If the recall score exceeds a threshold (0.5 in practice), we consider the reasoning path valid; otherwise, we discard the data point. In experiments where we sampled one CoT reasoning path per data point, this yielded a total of 17,895 CoT reasoning paths, of which only 375 (2.1%) were deemed invalid and discarded.

## B TRAINING DETAILS

For the best trained Confidence Reasoning Direct Optimization Models, we utilize Lora training on Llama-3-8B. The training data include TriviaQA, ConflictQA, NaturalQA, RedditQA. The training process was configured with a learning rate of 5e-6 and a maximum gradient norm of 0.3. The batch size per device was set to 1, with gradient accumulation over 4 steps, with distibuted on 4 A6000 GPUs. And the model was trained for 5 epochs. Additionally, 100 warmup steps were used, and the sequence length was constrained to a maximum of 900 tokens, with a prompt length limit of 600 tokens. For DPO, a beta value of 0.1 was applied, using a sigmoid loss function, while RPO was configured with an alpha value of 1.0. In terms of LoRA configuration, the model was set up with a LoRA alpha of 16, a dropout rate of 0.1, and a rank of 8 for parameter-efficient fine-tuning.

## C INTERNAL EVALUATION FINE-TUNING

We fine-tune the LLama-3-8B to do Internal Evaluation. We fine-tune it on three tasks including TriviaQA, RedditQA and NaturalQA. Where llama-3-8B first answer the training data and compare with the ground-truth to get the evaluation label. Then we fine-tune the model to do the evaluation. Evaluation Results are reported below

We observe that during InternalEval, the model's self-evaluation improves only on in-distribution tasks such as TriviaQA, ConflictQA, and NaturalQA, but performs poorly on OOD datasets like

| | TriviaQA | | | ConflictQA | | | NaturalQA | | | FreshQA | | | RedditQA | | | ClashEval | | |
|---|---|---|---|---|---|---|---|---|---|---|---|---|---|---|---|---|---|---|
| | *Internal Evaluation* | | | | | | | | | | | | | | | | | |
| | FR | TR | Acc | FR | TR | Acc | FR | TR | Acc | FR | TR | Acc | FR | TR | Acc | FR | TR | Acc |
| Llama-3 | 37.0 | 88.0 | 73.0 | 47.0 | 88.0 | 65.0 | 39.0 | 85.0 | 59.0 | 38.0 | 71.0 | 49.0 | 54.0 | 85.0 | 78.0 | 51.0 | 57.0 | 52.0 |
| Tuned | 76.0 | 77.0 | 77.0 | 89.0 | 57.0 | 76.0 | 89.0 | 56.0 | 75.0 | 97.0 | 38.0 | 77.0 | 100.0 | 13.0 | 33.0 | 99.0 | 12.0 | 21.0 |
| | *Situated Faithfulness with InternalEval Method* | | | | | | | | | | | | | | | | | |
| | TR | FA | OV | TR | FA | OV | TR | FA | OV | TR | FA | Overall | TR | FA | Overall | TR | FA | OV |
| Llama-3 | 79.0 | 61.7 | 70.4 | 69.0 | 40.0 | 54.5 | 60.3 | 38.7 | 49.5 | 55.7 | 24.0 | 39.9 | 84.7 | 66.5 | 75.6 | 53.3 | 16.7 | 35.0 |
| Tuned | 88.7 | 54.7 | 71.6 | 92.7 | 28.7 | 60.7 | 81.7 | 29.7 | 55.7 | 92.0 | 12.7 | 52.3 | 90.3 | 13.1 | 51.7 | 87.3 | 12.7 | 50.0 |

Table 5: Interval Evaluation Results: The top block presents results on Internal Evaluation, where FR (False Recall) indicates how effectively the model's wrong answers are identified by the internal evaluator, TR (True Recall) represents the recall for correct answers, and Acc reflects the internal evaluation accuracy. The bottom block shows results for situated faithfulness using the InternalEval method, comparing the performance of LLaMA-3 with LLaMA-3 fine-tuned using InternalEval. TR represents accuracy with the correct context, while FA represents accuracy with the wrong context.

RedditQA, and ClashEval. On these OOD datasets, the evaluation results shift from a bias toward evaluating answers as correct to incorrectly classifying them as wrong. This indicates that the InternalEval signal is noisy and lacks generalization. While the evaluation accuracy is low on datasets like ClashEval, the model's tendency to classify its own answers as incorrect results in an unexpected improvement in accuracy when given the correct document. This bias leads to an overall increase in Acc and the total score, highlighting a discrepancy between the model's internal evaluation ability and its situated faithfulness. This also underscores the limitations of rule-based confidence reasoning.

# D  FULL RESULTS

## D.1  FULL RESULTS FOR GPT-4O

Full results of GPT4-o are shown in Table 6.

## D.2  RCR'S CALIBRATION SCORES

To evaluate the calibration of confidence scores in the RCR methods, we present the calibration metrics—Expected Calibration Error (ECE) and AUC-ROC—for various models across different datasets, as shown in Table 7. LLMs demonstrate significantly better confidence estimation for their internal knowledge compared to external contexts. Additionally, while calibration improves the ECE score, this improvement does not necessarily translate to better situated faithfulness.

## D.3  RCR'S THRESHOLD SENSITIVITY

To demonstrate the sensitivity of RCR methods to threshold selection, we vary the threshold from 0 to 1 for the InternalConf method across different tasks on GPT-4o-mini. As shown in Figure 3, no universally optimal threshold applies to all tasks; instead, each task requires a tailored threshold for optimal performance. In contrast, SCR methods inherently bypass these threshold-tuning challenges, providing a more robust and consistent alternative.

# E  CONTEXT AND QUESTION ORDER EXPERIMENT

As shown in Table 8, in our preliminary experiments with GPT-4o-mini on incorrect contexts, we found that the language model tended to rely more on its internal knowledge when the question was presented before the context. However, when the context was presented first, the model was more easily misled

| GPT-4o-m | RedditQA | | | FreshQA | | | ClashEval | | | TriviaQA | | | PopQA | | | NaturalQA | | | Total |
|---|---|---|---|---|---|---|---|---|---|---|---|---|---|---|---|---|---|---|---|
| Method | TR | FA | OV | TR | FA | OV | TR | FA | OV | TR | FA | OV | TR | FA | OV | TR | FA | OV | OV |
| Closed-book | | | 80.7 | | | 53.0 | | | 20.0 | | | 81.7 | | | 56.7 | | | 62.0 | 59.0 |
| DIA | 96.0 | 12.5 | 54.3 | 96.3 | 2.3 | 49.3 | 85.3 | 12.0 | 48.7 | 96.0 | 12.3 | 54.2 | 97.7 | 11.0 | 54.4 | 88.7 | 10.3 | 49.5 | 51.7 |
| TACS (LR) | 93.8 | 16.5 | 55.2 | 86.3 | 4.6 | 45.5 | 76.3 | 14.3 | 45.3 | 92.3 | 16.0 | 54.2 | 76.3 | 14.3 | 45.3 | 86.0 | 15.7 | 50.9 | 49.4 |
| *Rule-based Confidence Reasoning (Evaluation Based)* | | | | | | | | | | | | | | | | | | | |
| ContextEval | 90.3 | 53.4 | 71.9 | 77.3 | 35.7 | 56.5 | 86.7 | 11.7 | 49.2 | 92.3 | 47.4 | 69.9 | 91.0 | 53.0 | 72.0 | 86.6 | 36.3 | 61.5 | 63.5 |
| InternalEval | 88.6 | 77.2 | 82.9 | 83.3 | 30.0 | 56.7 | 67.0 | 17.0 | 42.0 | 88.7 | 71.7 | 80.2 | 78.0 | 48.3 | 63.2 | 72.3 | 57.3 | 64.8 | 65.0 |
| *Rule-based Confidence Reasoning (Confidence Based)* | | | | | | | | | | | | | | | | | | | |
| TPC | 92.0 | 25.6 | 58.8 | 88.7 | 22.7 | 55.7 | 82.7 | 20.3 | 51.5 | 94.7 | 39.7 | 67.2 | 95.0 | 17.7 | 56.4 | 87.0 | 34.3 | 60.7 | 58.4 |
| CTPC | 90.3 | 56.3 | 73.3 | 80.0 | 28.0 | 54.0 | 67.7 | 23.0 | 45.4 | 93.0 | 47.7 | 70.4 | 82.7 | 40.7 | 61.7 | 82.0 | 42.0 | 62.0 | 61.1 |
| ContextConf | 93.8 | 13.1 | 53.5 | 82.3 | 16.3 | 49.3 | 80.0 | 16.7 | 48.4 | 93.0 | 19.7 | 56.4 | 94.0 | 14.3 | 54.2 | 85.0 | 30.6 | 57.8 | 53.2 |
| ContextConf(T) | 93.8 | 13.1 | 53.5 | 85.7 | 13.0 | 49.4 | 76.7 | 16.7 | 46.7 | 90.7 | 32.7 | 61.7 | 90.3 | 17.7 | 54.0 | 74.3 | 39.7 | 57.0 | 53.7 |
| ActiveRAG | 82.3 | 79.5 | 80.9 | 76.3 | 41.3 | 58.8 | 63.0 | 21.0 | 42.0 | 88.0 | 77.0 | 82.5 | 82.6 | 50.6 | 66.6 | 81.6 | 53.6 | **67.6** | 66.4 |
| InternalConf | 82.3 | 79.5 | 80.9 | 83.6 | 38.3 | 61.0 | 64.6 | 20.6 | 42.6 | 89.6 | 77.0 | **83.3** | 84.0 | 50.0 | 67.0 | 84.0 | 49.3 | 66.7 | 66.9 |
| InternalConf (T) | 83.5 | 79.5 | 81.5 | 86.0 | 36.3 | 61.2 | 75.6 | 21.0 | 48.3 | 89.0 | 78.0 | 83.5 | 83.6 | 51.0 | 67.3 | 85.0 | 44.6 | 64.8 | 67.8 |
| *Self-guided Confidence Reasoning* | | | | | | | | | | | | | | | | | | | |
| ImplicitSCR | 91.5 | 79.0 | **85.3** | 93.7 | 26.3 | 60.0 | 89.3 | 25.3 | **57.3** | 96.3 | 48.0 | 72.2 | 97.0 | 48.3 | **72.7** | 82.0 | 50.0 | 66.0 | 68.9 |
| ExplicitSCR | 92.0 | 73.9 | 83.0 | 82.7 | 47.0 | **64.9** | 82.0 | 17.0 | 49.5 | 92.3 | 71.3 | 81.8 | 93.0 | 51.0 | 72.0 | 80.3 | 51.0 | 65.7 | **69.5** |

| GPT-4o | RedditQA | | | FreshQA | | | ClashEval | | | TriviaQA | | | PopQA | | | NaturalQA | | | Total |
|---|---|---|---|---|---|---|---|---|---|---|---|---|---|---|---|---|---|---|---|
| Method | TR | FA | OV | TR | FA | OV | TR | FA | OV | TR | FA | OV | TR | FA | OV | TR | FA | OV | OV |
| Closed-book | | | 84.1 | | | 63.3 | | | 39.0 | | | 90.7 | | | 81.0 | | | 66.7 | 70.8 |
| DIA | 94.9 | 12.5 | 53.7 | 93.7 | 5.7 | 49.7 | 84.7 | 24.7 | 54.7 | 96.0 | 23.3 | 59.7 | 97.3 | 12.3 | 54.8 | 88.7 | 12.7 | 50.7 | 53.9 |
| TACS (LR) | 96.0 | 17.0 | 56.5 | 81.3 | 9.3 | 45.3 | 82.7 | 25.7 | 54.2 | 96.0 | 26.7 | 61.4 | 96.7 | 21.3 | 59.0 | 88.3 | 20.7 | 54.5 | 55.1 |
| *Rule-based Confidence Reasoning (Evaluation-based)* | | | | | | | | | | | | | | | | | | | |
| ContextEval | 92.6 | 66.5 | 79.6 | 87.7 | 44.0 | 65.9 | 81.3 | 23.0 | 52.2 | 94.3 | 71.3 | 82.8 | 96.7 | 73.3 | 85.0 | 86.7 | 51.3 | 69.0 | 72.4 |
| InternalEval | 87.5 | 81.3 | 84.4 | 81.3 | 40.0 | 60.7 | 76.7 | 30.0 | 53.4 | 95.0 | 85.7 | 90.4 | 88.7 | 70.3 | 79.5 | 74.3 | 60.3 | 67.3 | 72.6 |
| *Rule-based Confidence Reasoning (Confidence-based)* | | | | | | | | | | | | | | | | | | | |
| TPC | 94.3 | 71.6 | 83.0 | 87.0 | 39.7 | 63.4 | 77.0 | 39.0 | 58.0 | 94.3 | 75.3 | 84.8 | 96.7 | 66.7 | 81.7 | 84.0 | 50.3 | 67.2 | 73.0 |
| CTPC | 94.9 | 61.4 | 78.2 | 85.3 | 36.7 | 61.0 | 71.7 | 35.0 | 53.4 | 96.0 | 48.7 | 72.4 | 96.3 | 49.7 | 73.0 | 87.0 | 45.0 | 66.0 | 67.3 |
| ContextConf | 93.8 | 60.2 | 77.0 | 78.7 | 37.0 | 57.9 | 70.7 | 30.0 | 50.4 | 94.6 | 43.0 | 68.8 | 95.7 | 26.3 | 61.0 | 78.7 | 36.7 | 57.7 | 62.1 |
| ContextConf(T) | 92.0 | 68.8 | 80.4 | 71.0 | 48.0 | 59.5 | 75.0 | 30.3 | 52.7 | 92.0 | 63.0 | 77.5 | 93.0 | 35.7 | 64.4 | 71.0 | 57.7 | 64.4 | 66.5 |
| ActiveRAG | 89.7 | 78.9 | 84.3 | 82.0 | 51.7 | 66.9 | 59.3 | 40.6 | 50.0 | 93.0 | 88.0 | 90.5 | 88.0 | 79.0 | 83.5 | 77.0 | 61.0 | 69.0 | 74.0 |
| InternalConf | 89.8 | 78.9 | 84.4 | 83.7 | 50.0 | 66.9 | 61.6 | 40.6 | 51.1 | 93.0 | 88.0 | **90.5** | 88.7 | 78.7 | 83.7 | 78.6 | 59.6 | 69.1 | 74.3 |
| InternalConf (T) | 89.8 | 80.7 | 85.3 | 82.6 | 52.0 | 67.3 | 81.6 | 38.6 | 60.1 | 92.3 | 88.6 | 90.5 | 87.6 | 79.7 | 83.7 | 80.6 | 52.6 | 66.6 | 75.6 |
| *Self-guided Confidence Reasoning* | | | | | | | | | | | | | | | | | | | |
| ImplicitSCR | 92.6 | 75.6 | 84.1 | 82.0 | 54.0 | 68.0 | 86.0 | 44.3 | **65.2** | 94.7 | 85.7 | 90.2 | 96.0 | 76.7 | **86.4** | 83.3 | 66.7 | **75.0** | **78.1** |
| ExplicitSCR | 93.8 | 77.2 | **85.5** | 85.3 | 55.3 | **70.3** | 83.0 | 33.3 | 58.2 | 94.7 | 84.3 | 89.5 | 92.7 | 78.0 | 85.4 | 78.7 | 60.0 | 69.4 | 76.4 |

Table 6: Full Results for GPT-4o-mini and GPT-4o. "TR" represents accuracy given all correct contexts, "FA" represents accuracy given all wrong contexts, and "OV" represents the overall situated faithfulness, the main metric we focus on. The RCR method with grey background are the one which use an additional calibration set. ContextConf(T) means the threshold is tuned on the calibration set and it's the same for InternalConf(T). The best number among methods without additional calibration data is **bold** and the second best is underlined. The main findings are: 1) SCR methods generally outperform other methods across datasets 2) RCR methods also outperform basic baselines, but worse than SCR. 3) RCR can be improved with an additional calibraiton set, but still underperform SCR.

# F  MORE RELATED WORK

Sharing perhaps the most similar idea to ours, concurrent work AstuteRAG (Wang et al., 2024) proposes to iteratively consolidate internal and external knowledge while considering source reliability. Their major setup leverages actual results from a commercial search engine, which presents an arbitrary mixture of correct, incorrect, and irrelevant evidence in a single context. Focusing on a clean evaluation of LLMs' ability to handle a specific context, our setup controls the type of evidence the model sees at an instance level. We further explore various methods, including confidence-based, self-evaluation-based, and DPO methods, to elicit models' ability to resolve knowledge conflicts–all of which perform effectively. Another concurrent work, KnowPO (Zhang et al., 2024), also employs DPO to teach models how to resolve knowledge conflicts. However, their approach uses preference pairs based solely on final answers, while our CR-DPO method extends this by incorporating COT reasoning before the final answer. This not only enhances the model's reasoning ability but also provides explainability regarding its final decision. RobustRAG (Xiang et al., 2024) tackles potentially malicious documents by first generating LLM responses for each retrieved passage in isolation and then securely aggregating these responses. While their approach is designed for scenarios involving multiple retrieved documents, our focus is on a cleaner evaluation setting where the model must resolve knowledge conflicts with only a single passage at a time.

| | | TriviaQA | | | PopQA | | | NaturalQA | | | FreshQA | | | RedditQA | | | ClashEVal | | |
|---|---|---|---|---|---|---|---|---|---|---|---|---|---|---|---|---|---|---|---|
| | | ECE | AUR | SF | ECE | AUR | SF | ECE | AUR | SF | ECE | AUR | SF | ECE | AUR | SF | ECE | AUR | SF |
| llama-3 | IC | 13.6 | 78.6 | 73.8 | 23.9 | 80.5 | 58.5 | 24.2 | 70.2 | 51.3 | 24.4 | 76.2 | 46.5 | 18.4 | 66.8 | 77.3 | 23.6 | 74.7 | 44.7 |
| | CA | 6.8 | 78.1 | 71.7 | 5.9 | 80.0 | 62.5 | 2.6 | 69.3 | 53.5 | 8.4 | 76.2 | 51.0 | 11.0 | 66.7 | 76.7 | 3.7 | 73.4 | 49.7 |
| | CC | 40.8 | 50.3 | 55.3 | 41.8 | 49.2 | 55.8 | 35.5 | 54.5 | 48.0 | 32.5 | 57.8 | 42.5 | 43.1 | 54.9 | 56.9 | 32.8 | 56.7 | 41.7 |
| gpt-4o-m | IC | 7.7 | 82.1 | 83.3 | 11.7 | 83.0 | 67.0 | 15.6 | 76.4 | 66.7 | 11.6 | 78.6 | 61.0 | 15.8 | 87.4 | 80.9 | 25.9 | 76.1 | 42.6 |
| | CC | 37.9 | 52.4 | 56.4 | 41.9 | 43.2 | 54.2 | 30.4 | 56.7 | 57.8 | 35.1 | 50.2 | 49.3 | 45.9 | 48.1 | 53.5 | 36.5 | 61.8 | 48.4 |
| gpt-4o | IC | 6.5 | 84.3 | 90.5 | 9.4 | 86.2 | 83.7 | 18.2 | 74.2 | 69.1 | 14.2 | 80.0 | 66.9 | 10.1 | 71.8 | 84.4 | 27.6 | 83.8 | 51.1 |
| | CC | 36.3 | 46.4 | 68.8 | 39.9 | 41.5 | 61.0 | 30.0 | 57.0 | 57.7 | 27.4 | 56.1 | 57.9 | 32.4 | 63.7 | 77.0 | 25.0 | 56.4 | 50.4 |

Table 7: Calibration Scores for the RCR methods Across Different Models and Datasets: IC to the InternalConf, CA refers to InternalConf+Calibration with isotonic regression, and CC refers to ContextConf. ECE refers to Expected Calibration Error and AUR refers to AUC-ROC. SF refers to overall situated faithfulness. Results are with %. Findings are 1) LLMs demonstrate significantly better confidence estimation for their internal knowledge compared to external contexts. 2) while calibration improves the ECE score, this improvement does not necessarily translate to better situated faithfulness.

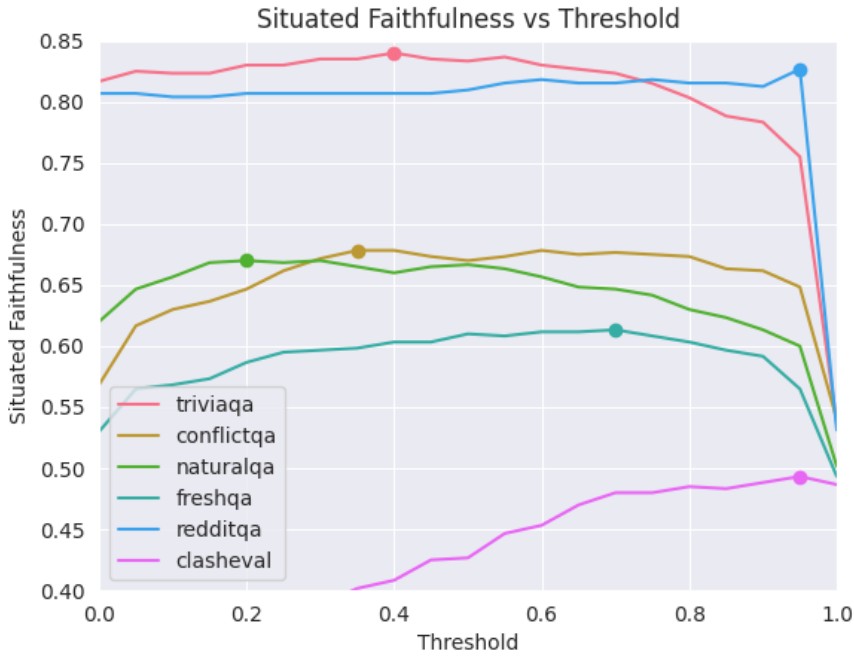

Figure 3: Impact of Thresholds on Situated Faithfulness for InternalConf Using GPT-4o-mini

While our work focuses on leveraging an LLM's own ability to decide whether to trust external knowledge, other approaches emphasize the use of additional retrieval metadata of external documents. For instance, Schlichtkrull (2024) explores source criticism of retrieved documents to help retrieval-augmented models answer controversial topics more effectively. Moreover, Credibility-Aware Generation (Pan et al., 2024) trains LLMs to adaptively utilize external documents based on their assigned credibility scores. However, such approaches face challenges due to the anonymity of many online sources and the way knowledge propagates through multiple intermediaries. Tracing information back to its original source and conducting a reliable credibility analysis can be difficult. Given these limitations, it is crucial to leverage an LLM's internal knowledge and reasoning ability to corroborate external information rather than relying solely on metadata-driven credibility assessments.

|     | TriviaQA | ConflictQA | RedditQA |
| --- | --- | --- | --- |
| QC  | 48 | 48.3 | 78.8 |
| CQ  | 33.3 | 33.1 | 46 |

Table 8: QC: effect, when question is put before context, model tends to utilize its own prior knowledge more. Results are acc given wrong contexts

## G  DISCUSSION

### G.1  SUMMARY

Our major takeaways include 1) LLM have the ability to resolve knowledge conflict, evidenced by large improvements of both RCR and SCR methods to direct input augmentation. 2) SCR outperforms RCR for strong models 3) CR-DPO could further enhance RCR. While the performance of ExplicitSCR and ImplicitSCR on specific tasks may depend on the dataset, we offer the following practitioner guidelines: 1) With a Development Set: If a development set is available, users can evaluate both ExplicitSCR and ImplicitSCR on the dev set to determine the best-performing method for their task. 2) Efficiency Priority: For tasks where computational efficiency is critical, ImplicitSCR is recommended due to its lower computational cost. 3) Explainability Priority: For tasks requiring interpretability, ExplicitSCR is the better option as it provides clear chain-of-thought reasoning.

### G.2  FUTURE WORK

Here we discuss several potential future directions for situated faithfulness. While our work primarily addresses knowledge conflicts in short-form QA, where answers are typically concise phrases, exploring knowledge conflicts in long-form QA is a promising avenue for future research. In long-form QA, internal knowledge may partially align with external contexts while simultaneously contradicting other aspects. This interplay introduces unique challenges in reconciling conflicting information and synthesizing coherent, accurate responses.

Additionally, our evaluation focuses on knowledge conflicts in controlled settings, where only a context is provided at a time. In RAG scenarios, there is significant potential for further exploration, particularly in cases involving multiple retrieved contexts. These contexts may not only conflict with internal knowledge but also contradict each other, presenting a more complex problem space.

Lastly, the CR-DPO framework requires additional computational resources during inference due to its reliance on COT reasoning. Future work could explore methods to distill the confidence reasoning process into a more efficient framework, retaining its effectiveness while reducing computational overhead.

## H  PROMPTS

Here are the prompts we use in the experiments. It should be noted that we use the "document" to represent the concept of context in real implementations.

### H.1  DIRECT INPUT AUGMENTATION

It utilizes 3-shot to help it format the answer.

> **Direct Input Augmentation**
>
> You will be given a question and a document. Utilize the information in the document to assist you in answering the question.
> Document: [Document]
> Question: [quesiton]
> Answer:

> **Direct Input Augmentation Examples**
>
> Example 1:
> Document: From the moment Sadie Frost and Jude Law met on the set of 1992 Brit flick, Shopping, she felt it was her destiny to "spend the rest of my life" with him. Married to Spandau Ballet star Gary Kemp, Sadie, then 25, tried to "crush her unwelcome ideas" about Jude, knowing they were "jeopardising an idyllic home life.
> Question: Who was married to Spandau Ballet's Gary Kemp and later to Jude Law?
> Answer: Sadie Frost
> Example 2:
> Document: Allegra Kent (CBA '19), ballerina and muse of George Balanchine and Joseph Cornell, started studying ballet at 11 with Bronislava Nijinska and Carmelita Maracci. In 1952, Balanchine invited her to New York City Ballet, where she danced for the next 30 years.
> Question: In which branch of the arts does Allegra Kent work?
> Answer: Ballet
> Example 3:
> Document:The magnificent tiger, Panthera tigris is a striped animal. It has a thick yellow coat of fur with dark stripes. The combination of grace, strength, agility and enormous power has earned the tiger its pride of place as the national animal of India.
> Question: Which animal is the national emblem of India?
> Answer: The Tiger

## H.2 IMPLICITSCR

> **ImplicitSCR**
>
> You will be given a question and a document. The document may not be trustworthy. Use your judgment to assess the reliability of the document. Then, based on both your assessment and your own knowledge, provide the best possible answer.
> Question: [quesiton]
> Document: [Document]
> Answer:

It utilizes the same 3-shot as Direct Input Augmentation to format the answer but order of the question and document follow the ImplicitSCR above template.

## H.3 EXPLICITSCR PROMPT

> **ExplicitSCR**
>
> Task Overview: You will be given a question along with your internal answer, a document that may contain either true or false information, and the document's answer to the same question. Your task is to evaluate the reliability of the document and determine whether the document is deceptive or not.
> Steps:
> 1.Internal Reasoning: Reflect on how you arrived at your internal answer using your own knowledge. Break down your reasoning process and assess the confidence level of your original answer, explaining why you believe your answer is correct.
> 2. Document Evaluation: Analyze the document and cross-reference the information provided with the known facts you used to form your internal answer. Determine whether the document contains deceptive or unreliable information, considering possible contradictions or inconsistencies.
> 3. Final Judgment: Based on your analysis, decide which answer (your internal answer or the document's answer) is more likely to be correct. Clearly state your final answer.
> Question: {question}
> Your answer: {internal answer}
> The document to judge: {doc}
> The document answer: {doc answer}
> Please provide a detailed reasoning process, followed by your final judgment. Ensure the last line of your response contains only the final answer without any additional explanation or details.

---

**ExplicitSCR Examples**

Example 1:

Question: What is the capital of France?

Your answer:Paris

The document to judge: London is the capital of France and the city stands as a vibrant symbol of French culture and governance. Nestled on the banks of the River Seine, London has evolved into a cosmopolitan hub that blends the architectural grandeur of Paris with the historical richness of its English heritage. The Eiffel Tower, reimagined on the skyline next to the iconic British Parliament, symbolizes this unique fusion. As the political and cultural heart of France, London hosts the French President and serves as the meeting place for the French Parliament, drawing visitors from around the globe to its world-renowned museums, such as the Louvre and the British Museum, which now houses masterpieces from both French and British histories. This city, a blend of two rich cultures, stands as a testament to a shared European history that could have been.

The document answer: London

I know from general knowledge that Paris is the capital of France. Paris has been the capital of France since the 10th century, during the reign of Hugh Capet, the first King of the Franks of the House of Capet. Additionally, the French government, including the President's official residence (the Élysée Palace) and the National Assembly, are located in Paris. Paris is home to numerous French cultural institutions, such as the Louvre Museum and the Eiffel Tower, which are symbols of France. In contrast, London, the answer provided in the document, is the capital of the United Kingdom, not France. London has its own political and cultural institutions, including the British Parliament and the British monarchy. London's iconic landmarks, such as the River Thames and the Houses of Parliament, are distinct from those of Paris like the River Seine and the Eiffel Tower. Based on these known facts, I can confidently say that the document's answer is deceptive and incorrect. , the final answer is:
Example 2:

Question: What is Paul Caillaud's occupation?

Your answer:Football player

The document to judge: Paul Caillaud (14 September 1917 – 15 August 2008) was a French pharmacist and politician. He represented the Independent Republicans (from 1962 to 1978) and the Union for French Democracy (from 1978 to 1981) in the National Assembly.[1] He was the mayor of La Roche-sur-Yon from 1961 to 1977.

The document answer: Pharmacist and politician

I initially thought Paul Caillaud was a football player. However, upon further reflection, I realized that I couldn't find any specific information about a person named Paul Caillaud. I don't know what football team he played for, and it is possible that he is a private individual or not well-known in public sources. The document claims that Paul Caillaud was a French pharmacist and politician, providing specific details about his political career, including his party affiliations and his role as the mayor of La Roche-sur-Yon. The document also includes his birth and death dates, which are consistent with a person's life history. Given the lack of information in my internal knowledge and the detailed information in the document, I believe the document is accurate., the final answer is:and politician

Example 3: Who won Super Bowl XX?

Question: What is Paul Caillaud's occupation?

Your answer:Chicago Bears

The document to judge: The New England Patriots won Super Bowl XX, defeating the Chicago Bears 46-10. The game was played on January 26, 1986, at the Louisiana Superdome in New Orleans, Louisiana. The Patriots' victory marked their first Super Bowl win in franchise history.
The document answer: New England Patriots
I recall that the Chicago Bears won Super Bowl XX, as they had a dominant season in 1985 and were known for their strong defense, ....

## H.4 INTERNAL EVAL

> **Filter context**
>
> Your task is to evaluate the model's response to a question. You will be provided with a question, the model's answer. Your job is to determine whether the model's answer is true or false.
> Question: [question]
> Model Answer: [model answer]
> Is the model's answer true or false?
> Return "True" if the model's answer is correct, and "False" if the model's answer is incorrect.

## H.5 CONTEXT EVAL

> **Context Eval**
>
> You will be given a question and a document that answers the question. Your task is to evaluate whether the document provides a correct answer to the question. If the document's answer is correct, return "True"; otherwise, return "False".
> Question: [question]
> Document: [doc]
> Is the document correct?
> Return "True" if the document's answer is correct, and "False" if the document's answer is incorrect.

## H.6 FILTER CONTEXT

> **Filter context**
>
> You will be given a document and a question. You need to remove the sentence which you think is not correct. You can only do removal and you can not add any new information or change the existing information. Only return the filtered document as your output.
> Here are some examples:
> Example 1:
> Document: The Eiffel Tower is located in Paris, France. It is the tallest structure in Paris. The Eiffel Tower was built in the 19th century and is made of wood.
> Question: Where is the Eiffel Tower located?
> Filtered Document: The Eiffel Tower is located in Paris, France. It is the tallest structure in Paris. The Eiffel Tower was built in the 19th century.
> Example 2:
> Document: Donald Trump is the President of the United States. He was elected in 2016 as a Democrat. He is the 45th President of the United States. Donald Trump was born in New York City.
> Question: Who is the President of the United States?
> Filtered Document: Donald Trump is the President of the United States. He was elected in 2016. He is the 45th President of the United States. Donald Trump was born in New York City.
> Example 3:
> Document: Taylor Swift is a famous singer. She was born in 1989. Taylor Swift has won multiple Grammy Awards. She is known for her country music.
> Question: When was Taylor Swift born?
> Filtered Document: Taylor Swift is a famous singer. She was born in 1989. Taylor Swift has won multiple Grammy Awards. She is known for her country music.
> Document: [doc]
> Question: [question]

## H.7 SAMPLING SELF CONFIDENCE REASONING

There are two different prompts here for sampling confidence reasoning paths. First, when the internal answer is correct and the context is wrong and second, when the internal answer is wrong but the context is correct. For the first case, we will first tell model its answer is correct and context is wrong and sample a chosen path, then we lie to the model its answer is wrong but context is correct and sample a rejected path. We do the same for the second case.

---

**sample for correct internal answer but wrong context**

You will be given a question, your answer to the question, a document and its answer to the question. Your answer is correct and the document's answer is deceptive. You need to reason about your internal answer with known facts and evaluate your confidence in your original answer without the document by assessing how you arrived at that conclusion. Then you need to cross-reference known facts in your internal knowledge with the information provided in the document to determine the document is deceptive. Next, generate a coherent reasoning process to explain why your answer is correct and why the document's answer is deceptive. Finally, provide the final answer based on your reasoning process. 'Example 1:
Question: What is the capital of France?
Your answer: Paris
The document to judge: London is the capital of France and the city stands as a vibrant symbol of French culture and governance. Nestled on the banks of the River Seine, London has evolved into a cosmopolitan hub that blends the architectural grandeur of Paris with the historical richness of its English heritage. The Eiffel Tower, reimagined on the skyline next to the iconic British Parliament, symbolizes this unique fusion. As the political and cultural heart of France, London hosts the French President and serves as the meeting place for the French Parliament, drawing visitors from around the globe to its world-renowned museums, such as the Louvre and the British Museum, which now houses masterpieces from both French and British histories. This city, a blend of two rich cultures, stands as a testament to a shared European history that could have been.
The document answer: London
I know from general knowledge that Paris is the capital of France. Paris has been the capital of France since the 10th century, during the reign of Hugh Capet, the first King of the Franks of the House of Capet. Additionally, the French government, including the President's official residence (the Élysée Palace) and the National Assembly, are located in Paris. Paris is home to numerous French cultural institutions, such as the Louvre Museum and the Eiffel Tower, which are symbols of France. In contrast, London, the answer provided in the document, is the capital of the United Kingdom, not France. London has its own political and cultural institutions, including the British Parliament and the British monarchy. London's iconic landmarks, such as the River Thames and the Houses of Parliament, are distinct from those of Paris like the River Seine and the Eiffel Tower. Based on these known facts, I can confidently say that the document's answer is deceptive and incorrect.
Therefore, the final answer is: Paris
Example 2:
Question: Who won Super Bowl XX?
Your answer: Chicago Bears
The document to judge: The New England Patriots won Super Bowl XX, defeating the Chicago Bears 46-10. The game was played on January 26, 1986, at the Louisiana Superdome in New Orleans, Louisiana. The Patriots' victory marked their first Super Bowl win in franchise history.
The document answer: New England Patriots
I recall that the Chicago Bears won Super Bowl XX, as they had a dominant season in 1985 and were known for their strong defense, led by players like William "The Refrigerator" Perry and Mike Singletary. The Bears' victory over the New England Patriots in Super Bowl XX was a significant moment in NFL history. The document claims that the New England Patriots won Super Bowl XX, which contradicts my internal knowledge. The document provides specific details about the game, including the final score and the location, but it fails to acknowledge the Chicago Bears' victory. Based on my knowledge and the inconsistencies in the document, I believe the document is deceptive.
Therefore, the final answer is: Chicago Bears
Now it's your turn.
Question: [question]
Internal answer: [model answer]
The document: [doc]
The document answer: [doc answer]
Now provide your reasoning process to explain why your internal answer is correct and why the document's answer is deceptive. Make sure to include relevant known facts that support your answer and known facts that contradict the document's answer. Your response should be coherent and logical, providing a clear explanation of your reasoning process and the final answer in the last line. Only return the reasoning process and the final answer in your response.

---

**sample for wrong internal answer but correct context**

You will be given a question, your answer to the question, a document and its answer to the question. Your answer is wrong and the document's answer is correct. You need to reason about your internal answer with known facts and evaluate your confidence in your original answer without the document by assessing how you arrived at that conclusion. Then you need to cross-reference known facts in your internal knowledge with the information provided in the document to determine to use the document to update your answer. Next, generate a coherent reasoning process to explain why your answer is wrong and why you choose the document answer. Finally, provide the final answer based on your reasoning process.

Question: When was the first robot used in surgery?

Your answer: 1995

The document to judge: The first robot used in surgery was the PUMA 560 robotic surgical arm, which was used in a neurosurgical biopsy in 1985. The PUMA 560 was developed by Victor Scheinman at Stanford University and was used to perform a minimally invasive brain biopsy. This groundbreaking procedure marked the beginning of robotic surgery and paved the way for future advancements in the field.

The document answer: 1985

I initially thought the first robot used in surgery was in 1995 because I remember reading about the da Vinci Surgical System, which was introduced in the 1990s and revolutionized surgical procedures. While I was aware of the da Vinci system, it seems too advanced to have been the first robot used in surgery, and there may have been earlier prototypes or systems that I'm not familiar with. The document provides specific information about the PUMA 560 robotic surgical arm, developed in 1985, which was used for a neurosurgical biopsy. A simple robotic arm like the PUMA 560 would be more likely to have been the first robot used in surgery, as it was a pioneering step in the field. Also, the 10 years gap between a robtic arm and a complex system like da Vinci is reasonable. Based on this new information, I realize that my initial answer was incorrect, and the document's answer is more accurate.

Therefore, the final answer is: 1985

Example 2:

Question: What is the largest catholic high school in america?

Your answer: Xavier High School

The document to judge: St. Francis Preparatory School, commonly known as St. Francis Prep, is a private, independent Catholic college preparatory school in Fresh Meadows, Queens, New York City, New York. It is the largest non-diocesan Catholic high school in the United States. St. Francis is run by the Franciscan Brothers of Brooklyn, who maintain a residence on the top floor of the school. As of the 2015–16 school year, enrollment at St. Francis was 2,489.

The document answer: St. Francis Preparatory School

My original answer was Xavier High School, which I believed to be the largest Catholic high school in America because the enrollment number was more than 1000 in 2020. However, the document provides information about St. Francis Preparatory School, which has a larger enrollment of 2,489 students as of the 2015–16 school year. The document also specifies that St. Francis Prep is the largest non-diocesan Catholic high school in the United States, which indicates its significant size and status. Based on this new information, I realize that my initial answer was incorrect, and the document's answer is more accurate.

Therefore, the final answer is: St. Francis Preparatory School

Now it's your turn.

Question: [question]

Internal answer: [model$_a$nswer]

The document: [doc]

The document answer: [doc answer]

Now provide your reasoning process to explain why your internal answer is wrong and why you choose the document answer. Make sure to include relevant known facts that contradict your answer and known facts that support the document's answer. Your response should be coherent and logical, providing a clear explanation of your reasoning process and the final answer in the last line. Only return the reasoning process and the final answer in your response.

## I    REDDITQA

### I.1    HUMAN ANNOTATION GUIDELINE

#### I.1.1    DATASET OVERVIEW

This dataset is specifically designed to evaluate the resilience of large language models against misleading information derived from real-world erroneous documents. It features a collection of datapoints, each consisting of a real-world, factually incorrect document and a corresponding factoid question crafted to highlight the document's inaccuracies. Note: A factoid question is a concise, self-contained query that demands a specific, definite, and verifiable piece of factual information, typically resolvable in a single sentence or a brief phrase. Verification of the answer should be possible through reputable sources, including authoritative websites (e.g., government websites, Wikipedia, education institutions, established news organizations), textbooks, peer-reviewed conferences or journal papers, widely accepted common sense, or confirmation from at least three secondary sources. For instance, a factoid question could be, "What is the tallest building in the world as of 2024?" This question is definite and verifiable. Conversely, a non-factoid question could be, "What is considered the best movie of all time?" This question is subjective and unverifiable, as it relies on personal opinions rather than objective facts.

#### I.1.2    COMPONENTS OF EACH DATAPOINT

**Document:** The document is a real-world text containing factual inaccuracies. These documents are selected for their relevance to common real-world topics and for the subtle nature of their inaccuracies, which might typically mislead both humans and AI models.
**Question:** Each datapoint includes a factoid question which is crafted to be answerable using information from the associated document or through general knowledge. The question should be designed to lead to an incorrect answer if based solely on the misleading information provided in the document.
**Answers Choices:** There are four answer options for each question: one correct answer, one answer derived from the document, and two plausible yet incorrect answers designed to increase the difficulty of the task.
**Correct Answer:** This is the verifiably accurate answer to the question, confirmed through reliable and independent sources. Serving as a gold standard, the correct answer allows for a precise evaluation of whether the model can identify and ignore the document's inaccuracies to reach a factually sound conclusion.
**Document Answer:** This answer is the most plausible but incorrect conclusion drawn from the document's misleading information. It serves as a test case for the model's susceptibility to being deceived by errors that might appear logical at first glance.

#### I.1.3    ANNOTATION PROCESS

Each datapoint you receive will include a document, a question, four potential answer choices, the correct answer, and the document answer (the answer according to the provided document). Your task is to confirm the validity of these components (details on what constitutes validity are provided below). If any element is invalid, modify it to ensure validity. If modification isn't possible, mark it as abstain. You need to fill in eight fields,

**Valid:** Whether the final datapoint is valid or not. Fill in yes if it's valid otherwise no.
**Question:** The final question (if the datapoint is not valid, skip 8)
**Answer Choices:** The four answer choices for the question (if the datapoint is not valid, leave it blank)
**Correct Answer:** The correct answer for the question (if the datapoint is not valid, leave it blank)
Document Answer: The answer derived from the document (if the datapoint is not valid, leave it blank)
**URL:** The correct answer for the question (if the datapoint is not valid, leave it blank)
**Evidence:** The correct evidence from the website / you draft that support the correct answer (if the datapoint is not valid, leave it blank)

**Explanation:** Explanation for directly using, modifying, or abstaining from the datapoint

### I.1.4 ANNOTATION DETAILS

**Validity and Incorrectness of the Document Answer Alignment with Document:** Confirm that the document answer, aligns with the inaccuracies presented in the document.
Yes: Continue to the next step.
No: Modify the document answer to better align with the inaccuracies in the document.

**Factual Incorrectness:** Ensure the document answer is factually incorrect by cross-referencing it with reliable sources via Google search or by identifying contradictions with common sense.
Yes (Document answer is incorrect): Continue to the next step.
No: Abstain from annotating the datapoint.

**Validity of the Question**

**Self-contained:** Ensure the question is a standalone real-world knowledge query that does not require additional context to be understood or answered. Yes: Continue to the next step.
No: Modify the question to include necessary information to make it self-contained or remove dependencies.
**Time Sensitivity:** Assess whether temporal details are crucial to the answer.
Yes (Time-sensitive): Include the necessary time constraints in the question.
No: Proceed with the annotation.
**Bias:** Determine if the question is subjective, implying there is no objective or universally recognized answer, and opinions may vary.
Yes (It's biased): Abstain from annotating the datapoint.
No: Continue to the next step.
**Verifiability:** Imagine if the question has an answer that can be definitive.
Yes: Continue to the next step.
No: Abstain from annotating the datapoint.

**Verifiability of the Correct Answer**
**Accuracy and Verifiability:** Confirm that the correct answer is factually accurate and can be substantiated through reliable sources, known facts, or logical analysis.
Yes: Continue to the next step. Record the URL of the support document in the filed URL. Save the paragraphs that substantiate the correct answers in the field Evidence If there is no direct paragraph that supports the correct answer, but it can be deduced through reasoning from existing facts, compose a paragraph yourself to document this reasoning
No: Modify the answer if possible; otherwise, abstain from using the datapoint.
**Appropriateness of Difficulty:** Ensure the correct answer is neither too obvious nor obscured by the incorrect alternatives.
Yes (Correct answer is appropriate): Continue to the next step
No (Correct answer is too obscured/obvious): Adjust the difficulty by either making the correct answer less obvious or the incorrect alternatives more plausible. For instance, the question "What percentage of people experience severe side effects from taking Accutane (isotretinoin)?" with the correct answer "It depends on the individual" is too vague. Instead, refine it to "Generally, less than 5%

**Evaluation of Other Candidate Answers**
**Incorrectness and Misleading Quality:** Verify that each alternative answer is both clearly incorrect and plausible, relying on general knowledge or logical fallacies.
Yes: Go to the final steps
No: Adjust the alternatives to be incorrect yet plausible and Finished

**Final steps:**
Fill in the field valid: If you choose to abstain from the datapoint, enter "no." If the datapoint is valid, enter "yes."
Fill in the field explanation: Provide an explanation for either abstaining from the datapoint or for directly using or modifying it.
Fill in the rest of the fields: If the datapoint is valid, complete the additional fields as required. If it is not valid, leave them blank.

**Additional Considerations** Whenever modifications are made to any component (question, document answer, correct answer), reassess the other components to ensure coherence and consistency across the datapoint. Only abstain from using the datapoint when modifications cannot resolve the issues.
The primary objective is to leverage as many real-world erroneous documents as possible for creating these factoid questions. Aim to modify questions and answers to make them workable within the guidelines provided. Exercise flexibility and judgment when necessary, abstaining only when essential.

### I.1.5 EXAMPLE 1:

Document:

It's not particularly humane, but ...

Boat people have made a choice. Go to a refugee camp (crappy choice), stay home and risk getting shot (crappy and dangerous choice), go on a boat to Australia (less crappy but dangerous and expensive choice). Unless Australia is the logical destination (Indonesian ethnic minorities?) then they shouldn't be encouraged to take a dangerous voyage to Australia.

If they go to a refugee camp, there's a good chance they will end up in a country that treats refugees like shit. Australia puts refugees (once they are processed) on a path to PR and citizenship, so they are almost as well-off as citizens. Places like Germany will treat them as second-class people indefinitely, unless they get deported.

If we want to really help people, we should accept more refugees from refugee camps (we do pretty well at this) or make refugee camps better. We could also push for better treatment of resettled refugees in other countries.

Permanent resettlement is not seen as an option by many countries. "You can come here for a few years, but should go home once you're no longer in danger". Many countries say it's better off if refugees just wait for the war to end, so they can go home. It's bullshit, and inhumane, but many countries take that stance. That's what we should be fighting, because Australia (and the other countries which treat resettled refugees like real people) can't fix all the worlds problems.

tl;dr - Australia does a good job permanently resettling 80,000 refugees a year (mostly from refugee camps, boat people are the worst choice because of the risk they die in a boat accident). Other countries should step up their game.

Before Annotation

Question: How many refugees does Australia permanently resettle annually, based on the latest data?

Answer Choices: A) 80,000 B) 13,750 to 18,750 C) 50,000 D) 100,000

Correct Answer: B) 13,750 to 18,750

Document Answer: A) 80,000

Annotation: Valid: Yes

Question: How many refugees did Australia permanently resettle annually between 2010-2020?

Answer Choices: A) 70,000-90,000 B) 10,000 to 20,000 C) less than 6000 D) more than 100,000

Correct Answer: B) 10,000 to 20,000

Document Answer: A) 70,000-90,000

Url: https://www.aph.gov.au/About_Parliament/Parliamentary_Departments/
Parliamentary_Library/pubs/BriefingBook47p/RefugeesAsylumSeekers

Evidence: The Austrilia Government sets a planning figure for the Humanitarian Program each year. The planning figure was around 13,750 for many years (with a temporary boost to 20,000 under the last Labor Government in 2012–13). Under the Coalition Government the planning figure reverted to 13,750, but was then gradually increased beginning in 2015–16 to provide for an additional 12,000 places over 4 years for people displaced by conflict in Syria and Iraq, before being reduced again to 13,750 in 2019–20. Table 1 shows total Humanitarian Program visas grants over the last 10 years, by offshore and onshore components.

• In the fiscal year 2011-12, the program distributed 6,718 grants for offshore resettlement and 7,038 for onshore protection, totaling 13,756 grants. • The following year, 2012-13, saw a significant increase in offshore resettlement grants to 12,515, with 7,507 onshore protection grants, bringing the total to 20,022. • The year 2013-14 reported 11,016 offshore and 2,752 onshore grants, totaling 13,768. • Similar figures were recorded in 2014-15, with 11,009 offshore and 2,750 onshore grants, summing up to 13,759. • In 2015-16, offshore resettlement grants jumped to 15,552, while onshore protection grants decreased to 2,003, totaling 17,555 grants. • The year 2016-17 marked the highest number of offshore resettlement grants at 20,257, coupled with 1,711 onshore grants, resulting in 21,968 total grants. • Offshore resettlement grants decreased to 14,825 in 2017-18, with 1,425 onshore protection grants, for a total of 16,250. • The number rose again in 2018-19, with 17,112 offshore and 1,650 onshore grants, totaling 18,762. • Similar onshore protection figures continued into 2019-20, with 1,650 grants, but offshore resettlement grants decreased to 11,521, resulting in a total of 13,171 grants. • The year 2020-21 saw the lowest numbers in the decade, with only 4,558 offshore resettlement grants and 1,389 onshore protection grants, totaling 5,947.

Example 2

Document: For the Pro, last time I checked, yeah - you can put any old 2.5" SSD in, or even replace the disk drive with a second HDD or SSD. The SSD in an Air probably can't be replaced, and if it could, you would need a very expensive PCI-E mounted drive. AFAIK, Samsung is the only company manufacturing such a drive for notebooks, and they aren't selling them retail, so Apple is one of the only companies in the world selling this hardware at any price. If you ignore the price of the 128gig drive you're upgrading from (and if you bought your own aftermarket replacement, you'd be eating the cost of that 128gig drive anyway): you can pay $200 to get a 256gig upgrade, or $500 to get a 512gig upgrade (it's a build-to-order option on the 256gig model for another $300). A quick browse through Newegg tells me that you'd be hard pressed to find a decent 256gig SSD, even a regular 2.5" drive, for under $200. You can get a decent 512gig drive for more like $400, so that's a bit of a markup, but only if you compare to the cheapest 2.5" drives money can buy. PCI-E is a lot faster than SATA, and these Air SSDs have about double the I/O throughput of those 2.5" drives. Even if you could just replace the Air's drive with a $400 512GB SSD, I think $100 is a small price to pay for substantially better performance, not to mention not having to do a complex, warranty-voiding installation process. A Google search for 512GB PCI-E SSDs found a Samsung replacement part for over $800, and a full sized desktop version from OCZ for over $2000. TL;DR: Apple is selling you new, super fast SSDs that you can't even buy at retailers yet, and they're charging barely more than the cheapest generic alternatives would cost for a normal, user replaceable SSD slot. Apple gets a bad rap for their past practices of gouging for HDD and RAM upgrades, but the Air storage upgrades are priced really well.

Before Annotation:

Question: Which of the following statements accurately reflects the upgrade options and pricing for SSDs in Apple's MacBook Air?

Answer Choices: A) Apple's SSD upgrades for the MacBook Air are priced similarly to the cheapest generic 2.5" SSDs available in the market. B) The MacBook Air uses standard, user-replaceable 2.5" SSDs, making upgrades straightforward and affordable. C) Apple exclusively uses Samsung-manufactured PCI-E SSDs for the MacBook Air, which are not available for retail purchase. D) MacBook Air's SSD upgrades are proprietary and generally priced higher than equivalent market rates for similar storage increases.

| | Human | Sentence | Modified | wiki-style | Anecdote | Reiteration | Average |
|---|---|---|---|---|---|---|---|
| DIA | 9.7 | 9.7 | 11.4 | 3.4 | 16.5 | 6.3 | 9.5±4.5 |
| RSCR | 43.8 | 49.4 | 41.5 | 34.7 | 46 | 46 | 43.6±5.1 |
| CRPO | 71.6 | 76.7 | 72.2 | 72.2 | 76.1 | 73 | 73.6±2.2 |

Table 9: Robustness to Incorrect Contexts: Each column represents a different type of incorrect context. The scores indicate the Untruthful Resistance (UR) for various methods. CR-DPO demonstrates the best robustness with both a high average UR and low variance across different context types.

Correct Answer: D) MacBook Air's SSD upgrades are proprietary and generally priced higher than equivalent market rates for similar storage increases.

Document Answer: A) Apple's SSD upgrades for the MacBook Air are priced similarly to the cheapest generic 2.5" SSDs available in the market.

After Annotation:

Valid: No

Question: [ ]

Answer Choices: [ ]

Correct Answer: [ ]

Document Answer: [ ]

Url: [ ]

Evidence: [ ]

Explanation: Abstain, The question is contingent on numerous variables, making the answers can't be verified The question has the following problems 1. Time-Sensitivity: Information regarding technology products, such as upgrade options and pricing, can change frequently due to market dynamics, promotional offers, and company policies. This means that what is accurate at one point may not be accurate later, making any single answer potentially obsolete soon after it is provided. 2. Lack of Specificity in the Question: The question does not specify a model year or a particular configuration of the MacBook Air, which can lead to multiple possible answers depending on the specific model and geographic market being considered. 3. Variable Regional Pricing: Apple, like many tech companies, often has different pricing and available options in different countries or regions. Therefore, the answer could vary significantly based on the geographic location. 4. Dynamic Market Conditions: Pricing and available options can also be influenced by temporary conditions such as supply chain issues, demand fluctuations, or promotional discounts, making a definitive answer difficult without specifying the exact timing and market conditions.

## J ROBUSTNESS OF CONFIDENCE REASONING PREFERENCE OPTIMIZATION

To assess the robustness of CR-DPO against various types of incorrect contexts, we conducted experiments on RedditQA, using different styles of wrong contexts, including:

**Human**: Incorrect human-written Reddit posts sourced from the internet.
**Sentence**: A single sentence containing an incorrect answer.
**Modified**: An LLM-modified version of the original correct context, subtly altered to lead to a wrong answer in a coherent way.
**Wiki-style**: An LLM-generated, wiki-style wrong document with made-up citations.
**Anecdote**: An LLM-generated anecdote where the wrong answer is mentioned.
**Reiteration**: The wrong answer is rephrased in 10 different sentences concatenated together .

As shown in Table 9, CR-DPO demonstrates greater robustness across different types of incorrect contexts, with a smaller variance in performance compared to other models. For instance, in the case of wiki-style wrong contexts—arguably the most misleading type for both direct input augmentation

and ESCR—CR-DPO still achieves a relatively high Untruthful Resistance (UR) score of 72.2%. Interestingly, for the "Sentence" context type, CR-DPO effectively resists the incorrect information. This suggests that CR-DPO is able to directly verify external facts against its internal knowledge, without being overly reliant on context features. This further implies that CR-DPO, during training, doesn't learn from superficial artifacts in incorrect contexts but rather learns to perform genuine fact-based confidence reasoning. Moreover, CR-DPO also handles "Reiteration" contexts, which involve repeated phrasing of the wrong answer—a context type known to be particularly misleading for RAG models, as highlighted in . CR-DPO's ability to defend against this type of misleading context aligns with prior findings, further validating its robustness.

