# OpenReview forum: "To Trust or Not to Trust? Enhancing Large Language Models' Situated Faithfulness to External Contexts"
_ICLR.cc/2025/Conference — ICLR 2025 Spotlight_

### Official Review · Reviewer_WXh7 · 2024-10-28

**Soundness:** 3
**Presentation:** 3
**Contribution:** 3
**Rating:** 5
**Confidence:** 3

**Summary:**

This paper uses confidence as a metric to adaptively determine when to rely on external knowledge and when to trust the model's parametric knowledge. The authors propose two pipelines: Self-Guided Confidence Reasoning (SCR) and Rule-Based Confidence Reasoning (RCR). Their findings suggest promising avenues for enhancing situated faithfulness in LLMs.

**Strengths:**

1. The presentation is clear, and the analysis is comprehensive. The authors conduct an in-depth examination of the situated faithfulness problem, covering a variety of QA benchmarks and models, including both closed-source models like GPT-4o and open-source models like LLaMA-3-8B. In addition to training-free methods, the authors also introduce a training-based approach, CR-DPO.
2. The authors address a significant problem: knowledge conflict, which is drawing increasing attention as large language models (LLMs) demonstrate greater capabilities. As LLMs evolve, effectively managing conflicting information between internal and external knowledge sources becomes increasingly critical.
3. The performance is satisfactory. Experiments on various benchmarks show both SCR and RCR has good performance compared with the baselines.

**Weaknesses:**

1. Limited Evaluation of baselines: While SCR and RCR are the primary contributions, the paper could benefit from a more extensive comparison to existing methods in RAG like Active RAG using confidence as a metric to adaptively decide when to use the external information. This omission impacts the novelty of the work.
2. Sensitivity to Confidence Thresholds: The effectiveness of some RCR methods hinges on predefined confidence thresholds, but the paper lacks a thorough analysis of threshold sensitivity across diverse tasks.
[1]. Active Retrieval Augmented Generation, Jiang et al. 2024

**Questions:**

Refer to the weakness.

---

> ### Author Response · Authors · 2024-11-19
> **Response to Reviewer WXh7 (Part 1 / 2)**
>
> Thank you for your time and effort in reviewing our work and for providing such insightful feedback.
> Regarding
> > Weakness 1: Limited Evaluation of Baselines, missing ActiveRAG
>
> Thank you for bringing up ActiveRAG. We acknowledge that ActiveRAG can be considered an instance within the broader scope of the RCR approaches. However, the presence of similar concepts does not undermine the novelty of our work.
>
> The core novelty of our contribution lies in **systematically formalizing** the SCR and RCR solution spaces under a **newly defined and important problem**, going beyond the scope of **any individual method**. Our work introduces a comprehensive framework that not only categorizes different approaches into SCR and RCR, but also critically analyzes their strengths and limitations, offering new perspectives and novel improvements. **This broader contribution, including formalization and framework-driven evaluation, extends beyond individual methods.** Consequently, the conceptual overlap of one single method within our framework with approaches like ActiveRAG does not detract from the significance of our work.
>
> We are also happy to provide a detailed analysis of how ActiveRAG fits within our framework and acknowledge its relevance more explicitly in the paper. Below, we discuss its connections and differences from our work from two perspectives
>
> - **Methodologies:** Their methodology could be categorized under RCR approaches. Under our setup, it will resemble the similar idea of InternalConf. The difference is in ways of confidence estimation – ActiveRAG utilizes external context when any one token’s probability is below a threshold,  while the InternalConf utilizes external context when the probability of the whole answer is below a threshold
> - **Motivations:** The problem setup and focus of ActiveRAG are fundamentally different from ours. ActiveRAG iteratively retrieves documents to refine model answers, aiming to enhance performance through external retrieval loops. It primarily focuses on correct or irrelevant contexts, neither accounting for incorrect contexts nor providing insights into the model’s true ability to resolve knowledge conflicts.  In contrast, our work evaluates the model’s ability to resolve knowledge conflicts between its internal knowledge and external contexts in a clean, controlled environment, focusing on model reliability and decision-making under these conditions.
>
> Results on GPT-4o and Llama-3 are shown below.  Overall, ActiveRAG’s performance is close to its conceptual counterpart, InternalConf, and does not impact our key findings:
>  1) LLM performs SCR better than RCR on strong models like GPT-4o
> 2) LLMs perform RCR better than SCR on smaller models like Llama-3-8B, while CR-DPO could enhance SCR.
>
>
> Thank you for this great suggestion and we have added the discussion of ActiveRAG into the part of InternalConf.
>
> | GPT-4o  | RedditQA | FreshQA | ClashEval | TriviaQA | PopQA| NaturalQA | Total |
> |------------|----------|----------|----------|----------|----------|----------|----------|
> | ActiveRAG     | 84.3     | 66.9     | 50.0     |  90.5    | 83.5     | 69.0     | 74.0     |
> | InternalConf   | 84.4     | 66.9     | 51.1     |  90.5     | 83.7     | 69.1     | 74.3     |
> | ImplicitSCR     | 84.1     | 68.0     | 65.2     |  90.2     | 86.4     | 75.0     | 78.1     |
>
> | Llama-3-8B | RedditQA | FreshQA | ClashEval | TriviaQA | PopQA | NaturalQA | Total |
> | :--- | :--- | :--- | :--- | :--- | :--- | :--- | :--- |
> | ActiveRAG | 77.3 | 45.2 | 43.3 | 73.7 | 56.5 | 51.0 | 57.8 |
> | InternalConf | 77.3 | 46.5 | 44.7 | 73.8 | 58.5 | 51.3 | 58.7 |
> | ExplicitSCR | 65.1 | 42.5 | 45.2 | 60.4 | 58.2 | 49.7 | 53.5 |
> | CR-DPO | 79.6 | 49.7 | 49.8 | 73.2 | 66.0 | 56.2 | 62.4 |

---

> ### Author Response · Authors · 2024-11-19
> **Response to Reviewer WXh7 (Part 2 / 2)**
>
> > Weakness 2: Sensitivity to Confidence Thresholds
>
> - **Sensitivity to RCR:** We agree that the RCR methods are sensitive to thresholds and no universally optimal threshold applies across all tasks for a single method. This aligns with our discussion in Section 5.3, where we highlight that the rule for RCR can be flawed (e.g. threshold is suboptimal), presenting a limitation of these methods.
>
> - **Mitigation through Threshold Tuning:** This limitation can be partially mitigated by tuning the threshold on a development dataset for each task, as demonstrated in Table 3. However, even with threshold tuning, RCR still underperforms compared to the SCR method (see Table 6) for GPT-4o, despite the advantage of leveraging additional data. This could be due to the noise in confidence estimations and the new bias of updated rules. SCR could avoid this pitfall by performing confidence reasoning internally in the text space.
>
> - **Additional Analysis Provided:** To provide a more comprehensive analysis as you suggest, we include plots illustrating how performance varies as thresholds range from 0 to 1 for each task, offering deeper insights into the behavior of RCR across tasks. As shown in Figure 3 in Appendix D.3, no universally optimal threshold applies to all tasks; instead, each task requires a tailored threshold for optimal performance. In contrast, SCR methods inherently avoid these threshold-tuning challenges, offering a more robust and consistent alternative.
>
> We appreciate the reviewer bringing up this point, as it reaffirms the value of our discussion on the limitations of RCR and our comparison with SCR as a more effective alternative.

---

> ### Author Response · Authors · 2024-11-23
> **Follow-Up on Addressed Reviewer Comments**
>
> Thank you again for your thoughtful comments and feedback. We sincerely hope that our response and revisions have addressed your concerns. If possible, could you please consider adjusting your scores in light of these updates? If you still have any remaining concerns or questions, please don’t hesitate to share them. We would be more than happy to discuss them further to ensure clarity and address any remaining issues.

---

### Official Review · Reviewer_2Etp · 2024-11-02

**Soundness:** 4
**Presentation:** 3
**Contribution:** 3
**Rating:** 8
**Confidence:** 3

**Summary:**

The paper explores how large language models (LLMs) handle internal knowledge versus external context, particularly when these sources conflict with each other. The authors argue that robust models should dynamically adjust their reliance on external context based on confidence scores in both internal knowledge and external context. To achieve this, they introduce self-guided confidence reasoning (SCR) and rule-based confidence reasoning (RCR) approaches and evaluate both open-source and close-source models across several QA datasets, including their own RedditQA, which they developed to study these behaviors. Results show that SCR is more effective for models with advanced reasoning abilities like gpt-4o, while RCR performs better for smaller models. They also propose Confidence Reasoning Direct Preference Optimization (CR-DPO) to create contrastive data and further enhances model performance.

**Strengths:**

Provides a comprehensive study on how LLMs manage situational faithfulness to external contexts. Very complete work with multiple approaches, new dataset and better solutions named CR-DPO.

**Weaknesses:**

While not critical, there are some minor writing flaws and citation issues, such as “model_answer” on page 20 and line 723 on page 14.

**Questions:**

When facing difficult problems, models sometimes fail to follow instructions and refuse to do so or produce illogical responses, especially when the model calibrates well. Have you done any check on the quality of data generated by CRDPO?  Specifically, how many samples were created, and what measures were used to ensure high data quality?

---

> ### Author Response · Authors · 2024-11-19
> **Response to Reviewer 2Etp**
>
> Thank you for your effort in reviewing our work and for the positive feedback—it is truly encouraging, and we deeply appreciate it.
>
> We also appreciate you pointing out the typo and citation issues, which we have now corrected.
>
> Regarding the quality of training data, we employ two key strategies to ensure robustness:
> 1. **Adding In-Context Examples:** We incorporate in-context examples when sampling COTreasoning paths from the model. These examples help guide the model to produce more coherent reasoning paths compared to using instructions alone.
> 2. **Post-Filtering:** To further enhance quality, we apply a post-filtering step to verify whether the ground-truth answer appears at the end of the selected CoT reasoning path. Specifically, we calculate the token-level recall score of the ground-truth answer within the last 3 $\times$ the length of the ground-truth answer tokens. If the recall score exceeds a threshold (0.5 in practice), we consider the reasoning path valid; otherwise, we discard the data point.
>
> We generated 5,000 data points each for TriviaQA, NaturalQA, and RedditQA, and 2,895 data points for PopQA. In experiments where we sampled one CoT reasoning path per data point, this yielded a total of 17,895 CoT reasoning paths, of which only 375 (2.1%) were deemed invalid and discarded.
>
> Thank you again for your suggestion. We have now included these details in Appendix A.2.

---

> > ### Comment · Reviewer_2Etp · 2024-11-26
> >
> > Thanks for the response and the revision of the paper! I found these information very helpful for practice. I'll keep my score.

---

### Official Review · Reviewer_UYZY · 2024-11-04

**Soundness:** 4
**Presentation:** 3
**Contribution:** 3
**Rating:** 8
**Confidence:** 3

**Summary:**

This work addresses the challenge of dealing with unreliable external contexts fed into LLMs. It introduces the concept of “situated faithfulness” suggesting that LLMs should evaluate whether to trust the external context or disregard it, answering the user’s question based solely on their parametric knowledge. To tackle this issue, two approaches are designed: Self-Guided Confidence Reasoning (SCR) and Rule-Based Confidence Reasoning (RCR). SCR prompts the LLM to assess the external context itself, while RCR uses predefined rules to determine the trustworthiness of the external context. The authors also propose a training method to enhance SCR, Confidence Reasoning Direct Preference Optimization (CR-DPO).
Experimental results validate the effectiveness of these methods, revealing several interesting findings.

**Strengths:**

- Situated faithfulness is a worth-investigating research problem, as knowledge conflict could occur in practical usage.
- The designed task RedditQA is a valuable resource for the community interested in this area. It would be beneficial if the author could open-source this dataset.
- The proposed methods are effective based on the conducted experiments.

**Weaknesses:**

- Both SCR and RCR introduce additional computational overhead and increase latency.

**Questions:**

- The authors apply multiple ways/prompts to estimate the confidence of the answers/context. I’m curious how calibrated these confidence scores are? Can you report the calibration metrics (AUROC, ECE) for these scores as well?
- Typos:
Line 022: self-access -> self-assess

---

> ### Author Response · Authors · 2024-11-19
> **Response to Reviewer UYZY**
>
> Thank you for your time and effort invested in reviewing our paper. We are grateful for your constructive comments and positive reception to our work, which has been really encouraging.
>
> Regarding
> > Weakness: Additional computational
>
> We agree that these methods may introduce additional computational overhead during inference. For RCR methods, generating both the internal answer and the context-based answer requires running the model twice. The computational overhead of the ExplicitSCR comes from the COT reasoning. Future research can potentially explore ways to distill the confidence reasoning process into a unified model. We have included a future work section, currently in the appendix to discuss this aspect.
>
>
> > Q1: calibration scores.
>
> Thanks for the great suggestion!  Below, we present the ECE scores and AUC-ROC scores for Llama-3 on TriviaQA (complete results for all models and tasks are included in Appendix D.2). The columns represent ECE, AUC-ROC, and overall situated faithfulness, while the rows correspond to the InternalConf, InternalConf+Calibration, and ContextConf methods.
> As shown in the table, the model demonstrates significantly better confidence estimation for its internal knowledge compared to external contexts. Additionally, while calibration improves the ECE score, this improvement does not necessarily translate to better situated faithfulness.
>
> Results are with %
> |  | ECE | AUR | SF |
> | :--- | :--- | :--- | :--- |
> | InternalConf | 13.6 | 78.6 | 73.8 |
> | InternalConf(Ca) | 6.8 | 78.1 | 71.7 |
> | ContextConf | 40.8 | 50.3 | 55.3 |
>
>
> > Q2: Typo.
>
> Thanks for pointing that out and we have modified that.

---

### Official Review · Reviewer_Z5rk · 2024-11-05

**Soundness:** 2
**Presentation:** 3
**Contribution:** 3
**Rating:** 8
**Confidence:** 3

**Summary:**

The authors propose multiple approaches for making large language models selectively reliant on either the context, or their parametric knowledge -- a framing termed situational faithfulness, where depending on the models' assessment of the truthfulness of context, it decides to use factual information presented therein, or alternatively, factual information encoded in the parameters.

SCR and RCR, two families of proposed approaches, rely on either soft assessment through different prompting and aggregation approaches, where the model yields the final output (SCR) or rule-based prediction based on the models' estimated confidence (RCR). The authors show that for advanced models (GPT-4o variants) SCR is the better approach, while for smaller models (LLaMA-3-8B), RCR prevails. Both of these families are better compared to baseline approaches across a number of QA tasks.
The authors also introduce a new QA dataset found "in-the-wild" in RedditQA, where questions and (correct as well as incorrect) answers were automatically generated based on claims flagged as uncertain or incorrect.

**Strengths:**

- The problem of balancing trust between contextual and parametric knowledge is important
- Comprehensive experimental setup and ablation study
- Well written and easy to understand
- Proposed methods outperform baselines and improve

**Weaknesses:**

- There is no real clear winner in terms of methods - in the case of GPT-4o, implicit and explicit SCR share the laurels between SCR methods, while InternalConf RCR is not far behind.
- The results are very densely presented and it is not clear what the takeaway is - using CR-DPO is the best approach if one can tune the model, however otherwise the instructions are very dataset dependent.

**Questions:**

None

---

> ### Author Response · Authors · 2024-11-19
> **Response to Reviewer Z5rk**
>
> Thanks for your insightful comments and positive review of our work, and we really appreciate that.
>
> Regarding the two weaknesses:
>
> We acknowledge that the performance of both ImplicitSCR and ExplicitSCR is close and the choice can be dataset-dependent. In this response, we aim to clarify the key takeaways, address concerns about dataset dependence, and provide practical guidance for real-world usage.
>
> - **Takeaways:** we want to emphasize our major high-level takeaways are: 1) LLM have the ability to resolve knowledge conflict, evidenced by large improvements of both RCR and SCR methods to direct input augmentation. 2) SCR outperforms RCR for strong models 3) CR-DPO could further enhance RCR
>
> - **Dataset-dependence:** We acknowledge that the best-performing method may vary depending on the dataset. However, we believe that averaging situated faithfulness across all datasets provides a reasonable indicator of a method’s general performance. As the diversity of datasets increases, it becomes less likely for a single method to excel universally. This approach aligns with standard practices in other benchmarks, such as MMLU, where overall results are compared despite domain-specific variations.
>
> - **Practical Usage Instructions:** To address real-world applicability, we have included practical guidelines for selecting a method:
> 1. If a development set is available, users can test ExplicitSCR and ImplicitSCR on the dev set to identify the best-performing method.
> 2. If efficiency is a priority, ImplicitSCR is recommended due to its lower computational cost.
> 3. If explainability is critical, ExplicitSCR is a better choice as it provides interpretable chain-of-thought reasoning.
>
> Thank you again for your thoughtful feedback. We hope the addition of usage instructions in Appendix F.1 further supports practitioners in applying these methods effectively.

---

> > ### Comment · Reviewer_Z5rk · 2024-11-27
> >
> > Thank you for the response, I believe the practical guidelines will strengthen the paper. I have adjusted my score.

---

### Meta-Review · Area_Chair_1EVi · 2024-12-25

**Metareview:**

This paper introduces the concept of “situated faithfulness,” where large language models (LLMs) dynamically determine whether to rely on ex ternal contexts or their internal knowledge based on confidence assessments. The authors propose two approaches—Self-Guided Confidence Reasoning (SCR) and Rule-Based Confidence Reasoning (RCR)—to enhance this capability. Results show that SCR outperforms RCR for advanced models, while RCR is more effective for smaller models, with both methods improving performance across QA datasets, including a newly introduced RedditQA dataset.

Strength: The paper addresses the critical issue of knowledge conflict in LLMs, presenting a clear and comprehensive study with well-designed methods and experiments. The introduction of the RedditQA dataset is a valuable contribution, and the proposed approaches demonstrate strong performance across benchmarks.

Weakness: Reviewers mentioned one potential weakness of the proposed methods is the additional computation overhead and latency. Also,  the effectiveness of SCR and RCR varies across datasets, making it challenging to identify a definitive best approach.

Overall, all reviewers acknowledge that the problem studied in this paper is quite important for the area, and the paper has made significant contribution.

**Additional Comments On Reviewer Discussion:**

The authors have addressed majority of the points raised by the reviewers in the rebuttal phase. Reviewer WXh7 still has some remaining concerns about one baseline method is performing comparable as one of the proposed method (RCR). However, given the paper has the additional contribution of systematically framing the situated faithfulness problem as well as performing systematic analysis, this should be a less concern overall. Therefore, I recommend acceptance of this paper.

---

### Decision · Program_Chairs · 2025-01-22

Accept (Spotlight)